
# The Potential of Smartstone Probes in Landslide Experiments: How to Read Motion Data

Bastian Dost[1], Oliver Gronz[1], Markus Casper[1], and Andreas Krein[2]

[1]Trier University, Campus II, Behringstraße, D-54296 Trier
[2]Luxembourg Institute of Science and Technology, Maison de l'Innovation, 5, avenue des Hauts-Fourneaux, L-4362 Esch-sur-Alzette

**Correspondence:** Bastian Dost (s6jodost@uni-trier.de)

**Abstract.** Currently, findings in landslide laboratory experiments are limited by observation techniques, which either deliver only external information (e. g. using high-speed videos), or internal information using wired sensors that confine the free motion of the mass. However, an unconfined internal observation of the internal dynamics of a moving landslide mass is essential for an adequate understanding of these natural hazards.

The present study introduces an autonomous and wireless probe to characterise motion features of single clasts within artificial laboratory-scale landslides. The Smartstone probe is based on an inertial measurement unit (IMU) and records acceleration and rotation at a sampling rate of 100 Hz. The recording ranges are ± 16 g (accelerometer) and ± 2000 ° s$^{-1}$ (gyroscope). The plastic tube housing is 55 mm long with a diameter of 10 mm. The probe is controlled and data is read out via active radio frequency identification (active RFID) technology. Due to this technique, the probe works under low-power conditions enabling the use of small button cell batteries and minimising its size.

Using the Smartstone probe, the motion of approx. 520 kg of an uniformly-graded pebble material was observed in a laboratory experiment. Single pebbles were equipped with probes and placed embedded and superficially in/on the mass. In a first analysis step, the data of one pebble is interpreted qualitatively, allowing for the determination of different transport modes, such as translation, rotation and saltation. In a second step, the motion was quantified my means of derived movement characteristics: The analysed pebble moved mainly in vertical direction during the first motion phase with a maximal vertical velocity of approx. 1.7 m s$^{-1}$. A strong acceleration peak of approx. 36 m s$^{-2}$ was interpreted as pronounced hit and led to a complex rotational motion pattern. In a third step, displacement was derived and amounts to approx. 1.1 m in vertical direction. The deviation compared to laser distance measurements was approx. -10 %. Furthermore, a full 3-dimensional spatiotemporal trajectory of the pebble was reconstructed and visualised supporting the interpretations. Finally, it is demonstrated that multiple pebbles can be analysed simultaneously within one experiment, allowing for motion sampling of different parts of a moving landslide.

## 1 Introduction

The spatiotemporal progression of moving slope material is subject of research in various geoscientific disciplines (e. g. Wang et al., 2018; Aaron and McDougall, 2019; Schilirò et al., 2019). Laboratory experiments are a well-established instrument





to investigate the physical behaviour of landslide motion processes. However, the observation of internal characteristics of a moving landslide mass poses a critical challenge. Nevertheless, an exact description of the internal behaviour is crucial to understand the mobility of these natural phenomena. The present study introduces an autonomous and wireless measuring device to observe the spatiotemporal motion of single clasts within a moving landslide mass in laboratory experiments.

## 1.1 Experimental investigation of landslide processes

To understand the physics of both dry and fluid-containing landslide processes on different scales and velocities, multitudinous experimental studies were undertaken during the last decades. For instance, Davies and McSaveney (1999) tried to reproduce large rock avalanches and concluded that some depositional features of very large granular avalanches may be caused by phenomena like rock fragmentation. Okura et al. (2000) conducted outdoor experiments to investigate the runout behaviour of rockfalls. They found that even though the centre of mass moved over shorter distances, the frontal part of the rockfall

body spread over a larger area. In addition, they observed by means of a visual particle tracking method that individual blocks did not change their relative positions during the motion precess. This means that frontal blocks were deposited in a distal zone. To explain these findings, Okura et al. argued that the frontal blocks gain additional dissipation energy because of clast collusions within the rockfall body. In contrast, rear blocks lost energy due to the collusions. Beyond that, Manzella and Labiouse (2009, 2013) investigated the influence of randomly or orderly stored blocks prior to the material release of artificial

granular landslides. These contrasting initial condition was used as an indicator for fragmentation. They found that the potential internal and external friction strongly influences the energy dissipation during the displacement process. For instance, if the bricks are stored randomly (high grade of fragmentation) or a sharp slope break exists (induces fragmentation), frictional and collisional conditions are pronounced and energy dissipation is intensified. In turn, this results in a strong spreading of the material.

These studies have in common that the displacing material is considered as one body changing its shape. Thereby, the motion process is observed from the outside and conclusions of the internal behaviour are drawn indirectly. This is a consequence of limited observation techniques. By means of (high-speed) video analysis such as particle image velocimetry (PIV) or the so-called fringe projection method (e. g. Manzella, 2008), only the surface or a 2-dimensional section of the body can be analysed. To overcome these restrictions, several methods were developed for the measurement of internal motion characteristics. For

instance, Yang et al. (2011) presented a detection system for impact pressure within debris flows and subsequently calculated the internal velocity. Additionally, wired devices such as piezometers, load cells and sensors for pore water pressure and deformation are common instrumentations for landslide experiments of various scales and objectives (e. g. Moriwaki et al., 2004; Ochiai et al., 2007; Ried et al., 2011).

Microelectronic devices for motion detection became common during the last years. Experimental studies use acceleration
sensors of Micro-Electro-Mechanical-Systems (MEMS) or combined acceleration- and rotation instruments such as inertial measurement units (IMU). After the early works of Ergenzinger et al. (1989) and Hanisch et al. (2003), who developed an intelligent boulder equipped with multiple sensors, sensor technology got more accessible and cheaper during the last decade. Several studies focused on technical aspects of so called 'smart tracers' for natural transport processes (e. g. Spazzapan et al.,



2004; Cameron, 2012). Others applied these techniques to geoscientific or geotechnical questions (e. g. Hofland et al., 2018). Volkwein and Klette (2014) presented a relatively large probe that could be embedded into boulders to record movement parameters of rockfalls. Although acceleration and rotation were recorded with high sampling rates to capture hard impacts of the rock, a further processing of the data was not carried out. The position of the rock during the displacement was tracked via wireless LAN. Another recent example of sensor techniques to describe gravitational induced movements is the Smart Soil Particle (SSP) presented by Ooi et al. (2014). Although acceleration data was interpreted quantitatively, a derivation of movement characteristics of the landslide motion was not performed. Another drawback of the SSP lays in the need of wires for energy supply and data transmission, which confines a free movement of the device within the soil.

The need of an autonomous and wireless device to investigate geomorphic transport processes was recently identified by Spreitzer et al. (2019). They presented a sensor based-probe to monitor the movement of artificial tree trunks during laboratory-scale flood experiments. Although a qualitative interpretation of the transport behaviour was done, a further processing of the data and a reconstruction of the trunks' trajectories were not carried out. Because under flood conditions, wood is mostly transported at the water table, the trajectory can be followed visually. In terms of landslide processes, this might not be possible. Here, it is of great importance to track material components that are embedded within the moving landslide body.

## 1.2 Scope of the present study

The present study introduces the Smartstone probe v2.0 as a device to measure movement characteristics of single clasts *in situ* within a surrounding mass. An experimental setup was developed that reproduces artificial landslides of a dry granular flow type. The experimental design focused on sensor application and not on natural landslide reproduction. The Smartstone probe is object of investigation in the present study. Photo/video documentation as well as reference measurements were carried out to verify the results (see Sect. 2). The present study deals with the following aspects:

1 A detailed description of the recent Smartstone probe prototype is given. Thereby, major changes to the former version are explained and the technical specifications are outlined.

2 An introduction of the additional information that are supplied by smart sensors is given. Specific properties of motion data are illustrated. Further, exemplary data patterns that emerged from the recordings are interpreted qualitatively. This includes an introduction on how to read motion data in terms of flume-scale landslide movements.

3 Subsequently, it is demonstrated how physical movement characteristics (see Sect. 3.2) can be derived from the raw measurements. The difference between sensor readings and movement characteristics are explained.

4 Another key aspect of the present study is to highlight the potentials of two- (2D-) and three- (3D-) dimensional visualisations of the path a clast took during the movement. These visualisation allows for an easy recognition of complex motion patterns.

5 Finally, a critical review of the limitations of the Smartstone probe prototype is carried out and future developments are outlined.



## 2 Material & methods

### 2.1 The Smartstone probe v2.0

In the present study, motion processes of single clasts were mainly observed by means of the Smartstone probe v2.0. The current prototype version is an improvement of the device that was presented by Gronz et al. (2016). A summary of the recent technical specifications is given in Table 1. All Smartstone kit components necessary to control the probe are shown in Fig. 1 (a). Contrary to the former version, which used a metal casing, the recent probe consists of an approx. 55 mm long and 10 mm wide plastic tube that holds the entire hardware. Therefore, the former external antenna could be replaced by an internal antenna, which allows an easier handling under experimental conditions. Energy is supplied by a single 1.5 V button cell battery (type AG 5). Two plastic plugs enable a waterproof closing. In standard configuration, the plugs have two sealing lips available. Under dry conditions, plugs with only one sealing lip can be used as well, reducing the probe's total length to approx. 50 mm.

Centrepiece of the probe is the approx. 30 mm long conductor plate holding an IMU with a combined accelerometer (ACC) and gyroscope (GYR) sensor – the Bosch BMI 160 (Bosch Sensortec GmbH, 2015). The 16 bit triaxial ACC measures accelerations ($a$) within the range of $\pm$ 16 g, which strongly enhances the recording range comparing $\pm$ 4 g of the former version ("g" as unit for gravitational acceleration). It exhibits a noise level of 1 mg at 100 Hz sampling rate. The 16 bit triaxial GYR measures rotations in terms of angular velocity ($\omega$) within the range of $\pm$ 2000 $° \, s^{-1}$ at a noise level of 0.04 $° \, s^{-1}$. The IMU is placed in the centre of the conductor plate. In addition, the probe is equipped with a magnetic sensor ('e-compass', MAG). For this purpose, the Bosch BMC 150 (Bosch Sensortec GmbH, 2014) is used to record within the range of $\pm$ 1300 µT (x-/y-axes) and $\pm$ 2500 µT (z-axis). Depending on the measuring range, the noise level is between 1 µT and 2 µT (the earth's magnetic field strength ranges between 22 µT and 67 µT). Sensor data and corresponding timestamps are stored on an internal 1 MB memory.

To allow an undisturbed motion, the Smartstone probe was developed as a wireless and autonomous instrument. The entire communication between the probe and a control software is performed by active radio frequency identification (active RFID) via the 868 MHz-band. Contrary to other communication techniques (such as wireless LAN), active RFID works under low power conditions enabling the use of small batteries for energy supply Additionally, it offers a higher operation range compared to Bluetooth. Because of this low-power communication technique the total size of the probe can be minimised. An USB-gateway works as interface between the probe and the controlling software with a graphical user interface (GUI). This enables the adaptation of probe settings, start of recording and data readout. For instance, a recording threshold can be set to avoid that minor signals (e. g. vibrations due to environmental perturbations) fill the internal memory before the considered motion begins. Moreover, single sensors can be switched off and the sampling rate can be adjusted (see Table 1). These settings will influence the time until the internal memory is completely filled. For instance, using all sensors (ACC, GYR, MAG) at a sampling rate of 100 Hz fills the memory in approx. 8 min of continuous measurement.

The probe dimensions of approx. 50 mm by 10 mm (minimal values) were chosen, because the probe should be embedded into representative clasts of the investigated material (see below). For this reason, a small button cell was used being aware that its capacity is limited. Yet, the Smartstone probe hardware could also be used to investigate the motion of lager objects. Hence,





longer plastic tubes and larger batteries (type AAAA) could be used for this propose. For the present study, five probes were
used, whereof one was damaged and could not be included into the analysis (see below).

The Smartstone probe v2.0 was developed and manufactured in cooperation with the company Smart Solutions Technology
GbR, Germany. Further improvements are planned and part of ongoing continuous research.

## 2.2   Axes conventions and reference systems

The following notations and conventions have to be considered during data description and interpretation. Due to the triaxial
architecture, sensor data is supplied by a triplet of values in each timestep. The triplet represents a vector with three space
components (Fig. 1, b). For instance, the ACC reading is composed of $a_x^p$, $a_y^p$ and $a_z^p$, where the subscript denotes the axis and
the superscript indicates that the values are probe readings (compare also Fig. 1, c). Note that ACC and GYR are mounted
on one side of the conductor board resulting in the same axis configurations. Contrary, the MAG is mounted on the opposite

side of the conductor board (rotated by 180 °). Therefore, its x- and y-axis are inverted. Following the right hand rule, positive
rotational directions are indicated by a small curved black arrow.

To compare the movement characteristics of different probes, the data must be displayed in higher-order reference systems
(1, c). The simplest way to do this, is the construction of a reference system using the probe's starting position as coordinate
origin. During the motion process, the probe follows a path (trajectory) that can be expressed as a time-dependent position

vector *relative* to the starting point. Therefore, the axes of this frame are donated with $x^{rel}$, $y^{rel}$ and $z^{rel}$. Note that the horizontal
trajectory component (x- and y-axes) of the probe and the relative reference system are identical for the first timestep. In
contrast, the deviation of the $z^p$-axis from the vertical direction can be determined exactly (for a detailed explanation see
Sect. 3.1). Therefore, it probably differs from the $z^{rel}$-axis. After the motion has started, the probe's orientation will change
while the relative reference system keeps its axes configuration. Consequently, within this reference system it is possible to

calculate the probe's orientation and the covered distance in each timestep. This enables both qualitative and quantitative
descriptions (see Sect. 3). In Fig. 1 (c) for instance, the probe has changed its orientation significantly compared to its starting
position while moving along the assumed trajectory.

However, a local reference system must be found to compare different probes with respect to the corresponding movement
characteristics and trajectories. For the present study, a local reference system relative to the experimental flume (see section

2.4) was defined. Following the former conventions, the axes were donated as $x^f$, $y^f$ and $z^f$. Note that the axes orientations of
the relative and the flume reference system may not be identical, except of $z^{rel}$ and $z^f$. This is due to the distinct deduction
of the vertical direction (direction of gravity) from ACC readings under stationary conditions (for a detailed explanation see
Sect. 3.1). Additionally, the local reference system ($x^f$, $y^f$, $z^f$) is not identical with a higher-order global system. To transform
local coordinates into a global system, the recording of magnetic data would be necessary, which was not done in the present

study.



## 2.3 Data calibration and processing

Prior to the calculations of movement characteristics, raw sensor data has to be calibrated. As further data processing uses ACC readings to derive movement characteristics, calibration is essential for the ACC. The recorded acceleration values of each axis ($a_x^p$, $a_y^p$, $a_z^p$) are generally erroneous due to three reasons: (i) a (quasi-) constant misreading, (ii) the imprecise orthogonal

alignment of the sensor axes, and (iii) the so called cross talk. Components (ii) and (iii) lead to the fact that a fraction of acceleration along an arbitrary axis will be measured on the two other axes. All of them can be corrected by adding a sensor- and environment depending vector (i) to the readings and multiply them with a scale factor matrix (ii and iii). Frosio et al. (2009) describes an optimisation algorithm that estimates these three components simultaneously. In the present study this approach was applied for the first time on Smartstone probe data. Because one probe was somehow damaged during the experiments,

ACC raw data of the remaining four probes was calibrated by means of the optimisation algorithm using MATLAB software. Subsequently, only these probes were analysed.

By means of the recorded acceleration and rotation data, the movement characteristics and the probe's trajectory can be reconstructed. Basically, these calculations are based on Newton's physical laws and integration of the recorded accelerations. Practically, if the pebble is in motion, gravitational acceleration and acceleration due to the motion will interfere. Nevertheless,

position and orientation in each timestep can be estimated by combining the ACC and GYR readings. This approach is termed *sensor fusion* (e. g. Koch, 2014).

In the present study a quaternion-based estimation algorithm was used that was originally developed to track the human gait. It was adapted from Madgwick et al. (2011) and x-io Technologies (2013) and supplies the movement characteristics velocity $v$ and displacement $s$ relative to the starting position. Additionally, it enables a 3D-visualisation of the trajectory. For a detailed

description of the computation see Sect. 3.2.

## 2.4 Experimental setup

The design of the experimental setup focussed on an exact and rapid triggering mechanism of the artificial landslide and flexibility for various studies. Figure 2 (a) shows the configuration that was used for the present study. A spring-based triggering mechanism allowed a rapid release of the material stored in a box on top of the flume. Eight single springs supplied a total

spring force of approx. 1660 N. After the release, the material moved along an approx. 4.2 m long plane inclined by 20 °. A small portion of the total material also entered the lower part of the flume that was inclined by 10 °. Lateral barriers limited the width of the flume to approx. 2.2 m. The bottom of the flume was covered by dimpled sheet to provide uniform basal frictional conditions.

A high-speed camera was placed close to the storage box to document the initial motion of the material. A camera of type

Optronis CR4000 x 2 and Tamron XR DiII (17-55 mm, 1:2.8) lens were used. High-speed sequences were recorded with 500 fps, a resolution of 2304 x 1720 pixels and were stored as *.jpeg-files. The camera was mounted with an inclination of 20 ° at the left side of the flume (direction of motion). The recorded pictures were mirrored during post-processing to achieve a better comparability between high-speed sequences and probe data. Therefore, motion proceeds from left to right in all





attached figures and the supplementary high-speed video (Video 1). The video facilitates the verification of the interpretation
(if the pebble is visible) of the sensor data and the concluded motion modes. We will refer to it several times.

For the present study, a uniformly-graded pebble material of fluvial origin (fluvial deposit of Moselle river) was used. Litho-
logically, it mainly consists of quartzite with smaller portions of greasy quartz and slate. Therefore, pebbles show laminated
and rounded to well-rounded shapes. The particle size range was specified to 32 mm to 64 mm and the effective unit weight
amounts to 1.55 t m$^{-3}$ (manufacturer information, EIDEN, 2017). A median particle diameter $d_{50}$ of 42 mm and a uniformity
coefficient $C_U$ of 2.1 was determined by sieving analysis. Clasts with diameters $\geqslant$ 60 mm amount to approx. 12 % (w/w) of
the material. A total mass of approx. 520 kg was used for the present study.

From the material several pebbles were taken to be equipped with Smartstone probes. For this purpose, a hole was drilled
through the pebble and modified in the way that a snug fit of the probe was achieved. Therefore, the probe could not move
within the hole during the motion process. Additionally, the pebbles were marked and numbered to be easily identified in
the high-speed sequences. The specific unit weight of each prepared pebble was determined by immersion weighing before
and after the preparation procedure. In this study, a detailed analysis of quarzitic pebble 4 will be carried out. By means of
immersion weighing a change in density (2.66 g cm$^{-3}$) was not detectable.

The storage box was filled with about 50 % of the material prior to the experiment. Two probe equipped pebbles were placed
and their position was measured at the temporal surface as displayed in Fig. 2 (b). Afterwards, more pebble material was filled
into the storage box and three more equipped pebbles were placed at the final surface. Figure 2 (c) and (d) show the initial
conditions prior to the experiment. The positions of each equipped pebbles were additionally measured relative to the upper
edge of the storage box. This was done by means of a laser distance meter. Measures were conducted for y$^f$- and z$^f$-direction
for both, the starting and the depositional position.

## 3   Motion data of landslide experiments and how to read it

The present study intends to demonstrate the motion behaviour of single pebbles that are transported within a moving mass
by means of IMU sensors. Exemplarily, one experiment, which was carried out in 2017, was chosen to present (i) sensor raw
recordings (Fig. 3), (ii) the derived movement characteristics ($a$, $v$, $s$, Fig. 4), and (iii) 2D- and 3D-visualisations (Fig. 5,
Fig. 6). The latter illustrate the complex motion trajectory of a single pebble within the landslide mass. Subsequently, data of
one pebble are analysed (sections 3.1 to 3.3) before the motion of multiple pebbles is considered in Sect. 4.1.

### 3.1   Qualitative description and interpretation of probe raw data

Fig. 3 shows the calibrated raw data of pebble 4. For this test, only acceleration in g (1 g = 9.81 m s$^{-2}$, Fig. 3, a) and rotation
in ° s$^{-1}$ (Fig. 3, c) were recorded (activated IMU). White bars indicate stationary (no motion) and black bars non-stationary
(motion) periods at the top of each plot. The previously explained data processing (see section 2.3) was only applied to non-
stationary periods. The whole motion sequence can be subdivided into six phases (A to F) with distinct properties characterising





a specific motion behaviour. Additionally, two discrete time points (diamond I and II) indicate major changes within the motion sequence. These phases and time markers highlight the same events in Fig. 3 to Fig. 5 and the supplementary video.

The motion sequence of pebble 4 covers a total duration of 2.1 s. Before the actual motion begins, $x^p$ and $y^p$ show very low values, though $x^p$ seems to be on a slightly higher level (approx. 0.0 g). At $y^p$, low negative values were recorded. Only at $z^p$ higher values of approx. 1 g can be seen. This pattern represents non-motion conditions, where only gravitational acceleration

is recorded. The plot of Fig. 3 (b) shows that the resultant acceleration $|\boldsymbol{a}|$ is approx. 1 g. According to the conventions from Sect. 2.2, $|\boldsymbol{a}|$ can be written as

$$|\boldsymbol{a}| = \left| \begin{pmatrix} a_x^p \\ a_y^p \\ a_z^p \end{pmatrix} \right| = \sqrt{a_x^{p\,2} + a_y^{p\,2} + a_z^{p\,2}} = 1g. \tag{1}$$

Each axis reading reflects a fraction of the gravity vector and is given by

$$a_x^p = \cos\alpha \cdot 1g; \quad a_y^p = \cos\beta \cdot 1g; \quad a_z^p = \cos\gamma \cdot 1g, \tag{2}$$

where $\alpha$ is the angle between $x^p$, $\beta$ is the angle between $y^p$, and $\gamma$ is the angle between $z^p$ and the gravity vector, respectively. Accordingly, under static conditions the probe's orientation relative to the gravity vector (downwards direction) can be calculated from the three readings of $a_x^p$, $a_y^p$ and $a_z^p$. Furthermore, the GYR data also shows that no rotation occurred before 0.0 s (Fig. 3, c), which also indicated stationary conditions.

**Phase A** (light yellow shading): The motion begins with a sudden change visible in all three plots. Between 0.0 s and approx.

0.03 s, a clear drop of $z^p$ to the halve of the former level is visible in the acceleration plot (Fig. 3, a). Simultaneously, the values of $x^p$ increase slightly above zero and those of $y^p$ slightly decrease. [1] Generally, relatively low acceleration readings are visible on all three axes during phase A, reflected by the resultant acceleration (Fig. 3, b). Low absolute values of acceleration can only be achieved if free fall is mixed with an additional component. Zero is only measured during pure free fall. Values between 0 and 1 imply a hampered free fall and / or an additional lateral acceleration.

In phase A the resultant acceleration is between zero and one. Hence, the pebble moved more or less downwards but was not free to fall. In fact, it was confined by the surrounding mass (see below). During phase A, angular velocities of about ± 250 ° s⁻¹ are visible in Fig. 3 (c). It is conspicuous that between 0.0 s and approx. 0.2 s, negative values are visible on $x^p$ and $y^p$, while $z^p$ shows positive values. Between approx. 0.2 s and 0.38 s, oppositional axes configurations with low absolute values at $z^p$ and positive angular velocities at $x^p$ and $y^p$ are displayed. These features show a forward- and backward rotation of

the pebble mainly around $x^p$ and $y^p$. Generally, phase A is characterised by relatively smooth curves without any large peaks and comparably low sensor readings for both, the ACC and the GYR. Thus, it appears that during this phase, a relatively calm

---

[1]The sign of the reading does not imply an increase or decrease of velocity. A positive value is caused by acceleration along this axis; a negative value is caused by acceleration in the opposite direction. A positive value as well as a negative value might be due to an increase of the pebble's velocity or a decrease – depending on its orientation.





motion behaviour was present without any stronger collisions between the pebble 4 and the surrounding clasts. One conclusion might be that the surrounding part of the mass moves coherently downwards.

**Diamond I and phase B** (light grey shading): At 0.389 s (diamond I) a distinct transition in the data sequence is visible.
Contrary to phase A, uneven and peaky curves can be seen in all plots. In Fig. 3 (a), $z^p$ generally shows high acceleration peaks of approx. 3.0 g. From 0.389 s to approx. 0.7 s, $x^p$ and $y^p$ show values of about + 1 g and – 1 g, respectively. The resultant acceleration (Fig. 3, b) also shows a peaky curve with values between 0.2 g and approx. 3.0 g. Looking at the GYR data, high angular velocities of about 600 ° s$^{-1}$ at $x^p$ and $y^p$ are visible around diamond I. This indicates a strong rotation around these axes and may be a hint for major changes in direction. After that, relatively low $\omega$ values $< 500$ ° s$^{-1}$ are recorded during phase B.

**Diamond II and phase C** (light yellow shading): At diamond II another strong transition is visible in the time series. The strongest peak of the whole sequence (approx. 4.6 g) is measured at $z^p$. Accelerations with this magnitude were actually recorded in two timesteps (0.898 s and 0.908 s). This is conspicuous since most of the other peaks consist of only one data point. Because of the low acceleration recordings of $x^p$ and $y^p$, the resultant acceleration is calculated to approx. 4.7 g. Diamond II introduces phase C, where higher sensor reading in GYR data are visible as well (Fig. 3, c). Here, the phase begins with
relatively low $\omega$ of approx. 260 ° s$^{-1}$ at 0.898 s. After that, a strong increase on $y^p$ is visible until at 0.918 s, a local maximum of approx. 1230 ° s$^{-1}$ is reached. Interestingly, this peak was recorded after high values were recognised at $a_z^p$, 0.01 s earlier. While the GYR readings of $x^p$ and $z^p$ are relatively low at approx. - 150 ° s$^{-1}$, values of $y^p$ stay at a high level of approx. 750 ° s$^{-1}$. At the end of phase C, an increase of $\omega$ at $y^p$ is visible.

These recordings can be interpreted in the way that pebble 4 changes its mode from lateral sliding to rotation and saltation.
This point in time is also clearly visible in Video 1 at the position marked with diamond II. In the following, each saltation is characterised by single strong peaks on different axes (as the pebble also rotates).

**Phase D** (light red shading): The short period between 1.008 s and 1.038 s (4 data samples) can be easily identified within the acceleration plots (Fig. 3, a and b). Low ACC readings of all three probe axes lead to a resultant $a$ close to zero. As explained above, this is only possible under almost free fall conditions. Therefore, it can be reasoned that the pebble 4 fell for approx.
0.03 s. The gyroscope plot (Fig. 3, c) shows again high values of approx. 900 ° s$^{-1}$ for $y^p$ and relatively low values for $x^p$ and $z^p$. This implies a pronounced rotation while the pebble fell.

**Phase E** (light yellow shading): A strong rotation around $y^p$ continues at the beginning of phase E. But contrary to the former phases, $\omega_x^p$ and $\omega_z^p$ show increasing positive and negative values since approx. 1.07 s, respectively. At approx. 1.14 s a peak of $\omega$ of approx. - 820 ° s$^{-1}$ occurs at $z^p$ before the values decrease again. At about the same point in time, strong peaks are
visible at the ACC readings at each probe axes. These lead to the second highest $a$ resultant (approx. 4.2 g) of the whole time series. From approx. 1.23 s to approx. 1.24 s another short period of ACC readings around zero is visible, resulting in an $a$ resultant of approx. 0 g. At the end of phase E, a last strong $a$ peak (3.6 g) at $z^p$ and a strong decline of the $\omega_y^p$ are visible. This denotes a major change in motion behaviour with a transition from strong rotations in the phases C, D and E to less rotational but translational displacement.

**Phase F** (light grey) and the end of motion: During this last phase, a continuous decline of $\omega$ at all probe axes can be seen. Whereas values of approx. ± 200 ° s$^{-1}$ are recorded at approx. 1.3 s, until the end of the movement an almost logarithmic





decrease of these values is visible. This decline appears also at the ACC readings from approx. 1.53 s onwards. At 1.826 s the end of the motion sequence is reached. GYR readings around 0 ° s$^{-1}$ were recorded. At the ACC, only minor changes can be seen after this point in time. At z$^p$ values vary slightly below 1 g. Readings of x$^p$ and y$^p$ are slightly higher than 0 g. As the

pebble is stationary, only the gravitational acceleration vector is displayed by the data. This is also visible in Fig. 3 (b), where the calculated $a$ magnitude varies around 1 g.

Concerning the whole time series, some interesting aspects shall be mentioned: The small deviations from the mean axes readings of the ACC after the motion (right white bar) can be interpreted as either vibration of the flume construction or surrounding clasts that are still in motion and, therefore, induce smaller accelerations to pebble 4. Of cause, an overlay of these

effects could also explain these recordings. Anyhow, these readings are too small to be induced by motion and are therefore excluded from further analysis. The same applies to the rotation data.

By comparing the ACC readings before and after the movement (white bars), a minor change of x$^p$ and y$^p$ can be seen. While x$^p$ showed values of ± 0.0 g and y$^p$ slightly negative readings before the start, low positive values were recorded after the motion on both axes. Contrary to this, z$^p$ shows slightly lower values after the motion compared to its readings before the

start of the experiment. From this can be reasoned that the orientation of pebble 4 after the movement has changed. Because the ACC readings of z$^p$ are slightly lower it follows that this axis does not point exactly into vertical direction after the motion and pebble 4 is somewhat tilted.

Further, different 'modes' of sensor readings occur during the motion sequence. The first mode is generally characterised by little ACC readings on all axes. In addition, the curves are relatively smooth and less peaky, which is particularly clear

for the rotation data. This mode is present in phase A and for the short period of phase D. The second mode consists of peaky and relatively high acceleration values simultaneously with relatively low, but peaky GYR readings. The amplitude of ACC values is relatively high. This mode occurs during phases B and F. Contrary, a third mode shows smoother (less peaky) ACC readings with lower amplitudes and high, but less peaky, GYR recordings. This mode can be observed in phases C and E. These oppositional observations reflect the previously mentioned motion behaviour. The first mode is recorded when the pebble

mainly falls downwards and clast contact is inhibited. The second mode is recorded if translational transport under confined conditions occurs. Pebble 4 moves within the mass and is exposed to pronounced collisional contacts due to surrounding pebbles. This results in frequent impacts and, consequently, acceleration peaks. Because the pebble is generally not free to move, larger rotation is inhibited and minor but sudden orientation changes occur. This is reflected by the relatively low but peaky GYR readings. Contrary, mode three occurs when the pebble rotates unconfined. This is only possible, while the pebble

is not surrounded by other material. This means that the pebble must be above the moving mass. In other words and geo-scientifically speaking: the pebble saltates. A alternating pattern of high $a$ peaks and almost zero acceleration magnitudes also fits to this process, because during saltation the pebble bounces at the flume bottom before it rebounds and falls again.

## 3.2   Quantifying motion by means of derived movements characteristics

The previously explained data only focused on the motion mode. Now, we want to investigate the movement with respect to

position and time. The recorded data is only a result of external influences (forces) that act on the pebble. However, from the


recorded and calibrated raw data, the pebble's movement characteristics relative to its staring position ($\boldsymbol{a^{rel}}$, $\boldsymbol{v^{rel}}$, $\boldsymbol{s^{rel}}$) can be derived by simple physical relations. The initial orientation of the pebble can be calculated after Eq. 1 and Eq. 2. By means of the received Euler angles $\alpha$, $\beta$ and $\gamma$, the sensor readings $a_x^p$, $a_y^p$ and $a_z^p$ can be rearranged to $a_x^{rel}$, $a_y^{rel}$ and $a_z^{rel}$ (compare also Sect. 2.2). However, the representation by Euler angles may not be bijective and therefore may lead to an erroneous

initial orientation ('gimbal lock'). Another method to derive initial orientation by means of acceleration and rotation data was presented by Madgwick et al. (2011). It is based on a quaternion representation and supplies bijective solutions (for detailed explanations the reader is referred to Madgwick et al. (2011) and specific literature as e. g. (Jazar, 2011)). It was implemented into a MATLAB algorithm, which was published online under the URL: https://x-io.co.uk/gait-tracking-with-x-imu/ (CC license, x-io Technologies, 2013).

After finding the initial orientation, the vector $\boldsymbol{a^{rel}}$ consequently gives the translational acceleration of the pebble within a reference system relative to the pebble's staring position (compare Fig. 1, c). Thereby, the direction of $a_z^{rel}$ equals the gravity vector and thus points downwards. Hence, $a_x^{rel}$ and $a_y^{rel}$ give the horizontal component of $\boldsymbol{a^{rel}}$. After the rearrangement of the recorded accelerations and with respect to time $t$, the movement characteristics $\boldsymbol{v^{rel}}$ and $\boldsymbol{s^{rel}}$ can be integrated as

$$\boldsymbol{v^{rel}}(t) = \int \boldsymbol{a^{rel}}(t)dt \tag{3}$$

and

$$\boldsymbol{s^{rel}}(t) = \int \boldsymbol{v^{rel}}(t)dt. \tag{4}$$

By applying these formula, movement characteristics were calculated for the non-stationary period and are plotted in Fig. 4 (a-c). Individual phases and distinct points in time are indicated in the same way as displayed in Fig. 3. Additionally, captures of the high-speed sequence from diamonds I and II are shown in Fig. 4 (d). Note also, that acceleration values are

plotted in the unit m s$^{-2}$. Data processing was applied from the start of motion (compare black bars in Fig. 3 to 5). Only $\boldsymbol{a^{rel}}$ was rearranged before the motion starts (white bars). Note that the values are defective during these stationary periods.

Relatively low acceleration values are calculated during phase A. As displayed in Fig. 4 (a), $a_z^{rel}$ increases continuously until at approx. 0.32 s a local maximum of approx. - 9.4 m s$^{-2}$ occurs. This is less than the gravitational acceleration (9.81 m s$^{-2}$). Therefore, it can be reasoned that free fall conditions were not totally developed during this phase. In fact, pebble 4 was confined

by the underlying mass. This can also be seen in Fig. 4 (d), where pebble 4 'swims' at the surface of the moving material. Therefore, phase A could be termed as 'confined fall'. The highest derived velocity of $v_y^{rel}$ during phase A was calculated to approx. 1.7 m s$^{-1}$ at 0.379 s. Afterwards the $v_z^{rel}$ velocity component decreased. Simultaneously, the $v_z^{rel}$ velocity component increased further. During phase A, a cumulated vertical distance of approx. 0.35 m was covered. The $s_y^{rel}$ component amounts to approx. 0.11 m at the end of phase A.

At 0.389 s after the start, a major change is visible in all plots. The corresponding capture of the high-speed sequence is shown in Fig. 4 (d). The relatively smooth acceleration curves change to a more peaky pattern. This pattern was already identified in Fig. 3. A first strong peak of approx. 14.7 m s$^{-2}$ occurred at $a_z^{rel}$ and marks the begin of phase B. At this time, a





transition from confined fall to translational movement occurs. Additionally, the peaky pattern of the acceleration and velocity
curves indicates pronounced clast contact and frictional behaviour. This is particularly clear for $v_z^{rel}$. Because pebble 4 moves

at the surface of the material, clast contact occurs mainly in vertical direction. During phase B, the vertical velocity component
subsequently decreases. Meanwhile, $v_y^{rel}$ increases until at 0.609 s the maximum of approx. 1.45 m s$^{-1}$ is reached.

Phase C again is introduced by a sudden strong increase in $a_z^{rel}$ at 0.898 s (diamond II). The acceleration peak at 0.908 s of
approx. 35.9 m s$^{-2}$ leads to a positive vertical velocity of approx. 0.21 m s$^{-1}$. The pebble consequently moves upwards at this
point in time, which can be seen in the displacement plot (Fig. 4, c). Afterwards, the displacement tends again to downwards

motion in phases D and E. During phases C to E, the displacement plot (Fig. 4, d) shows stair-like features at the $s_x^{rel}$ and $s_z^{rel}$
curves. These features can only be achieved if the actual motion acts against the tendency of downwards movement parallel
to the flume bottom (see Fig. 1). Together with the previously mentioned high $\omega$ around all probe axes (compare Fig. 3), a
complex rotational motion pattern can be interpreted until 1.307 s. Note that only the first milliseconds of this complex motion
are visible in Video 1 since pebble 4 left the field of view at approx. 1.0 s.

In phase F, translational acceleration and derived velocity components gradually decline, which leads to only little displace-
ments. A total displacement of approx. 1.0 m in z$^{rel}$-direction and 1.3 m in y$^{rel}$-direction was calculated by means of the former
mentioned algorithm. Additionally, in Fig. 4 (c) the covered distance measured with a laser distance meter in flume direction
are plotted. Although being aware that y$^{rel}$ and y$^f$ do not necessarily have to be identical (compare Fig. 1 and Sect. 2.2), a high
agreement between sensor-derived and manually measured displacements is displayed. It can be reasoned that the probe must

be oriented more or less in flume direction, following that the probe axes x$^{rel}$ and y$^{rel}$ $\approx$ x$^f$ and y$^f$. As the vertical direction
is derived from ACC readings under stationary conditions z$^{rel}$ equals z$^f$. Thus, the deviation between $s_z^{rel}$ and $s_z^f$ reflects the
quality of sensor-derived position. Whereas a sensor-derived vertical displacement of 0.999 m was calculated, a true vertical
displacement of 1.109 m was measured in fact. This means the calculations underestimate the vertical displacement by less
than 10 %.

## 3.3   Visualising motion by trajectory reconstructions

As described in Sect. 1, high-speed video recording is one of the traditional methods to observe rapid movements. Such a
video sequence was recorded for the present study as well (Video 1). Due to narrow conditions at the experimental facility,
the high-speed camera had to be installed very close to the setup, resulting in a relatively small field of view. At the end of
the high-speed sequence, nevertheless the start of a complex rotational motion of pebble 4 can be observed. However, the full

motion feature is not visible.

Although only the first portion of this complex motion is visible on the high-speed sequence, the full trajectory can be
reconstructed by means of the recorded Smartstone data. The trajectory is defined as the position vector composed of $s_x^{rel}$,
$s_y^{rel}$ and $s_z^{rel}$ for each timestep (Fig. 4, c). As additional information, the pebble's orientation can be reconstructed by means
of the previously described algorithm. Consequently, these variables can be plotted as a function of time within a Cartesian

coordinate system, as displayed in Fig. 5. Thereby, the axes x$^{rel}$, y$^{rel}$ and z$^{rel}$ donate the distance axes relative to the starting
position of the pebble. Note that z$^{rel}$ always points into vertical direction (for explanation see above) and that diagram axes of





Fig. 5 are not drawn in the same scale. Note that contrary to Fig. 3 and 4, Fig. 5 shows no time series but visualises the pebble's position within the relative reference system.

In Fig. 5 (a), the trajectory projected on the $y^{rel}z^{rel}$-plane can be seen, representing the same perspective as Video 1. Addition-
ally to the side view perspective, data can also be visualised as top view, where the trajectory is projected on the $x^{rel}y^{rel}$-plane (Fig. 5, b). Moreover, the 3D-trajectory can be visualised as displayed in Fig. 5 (c). The pebble's position is marked by small black dots and probe's axes are shown in red ($x^p$), green ($y^p$) and blue ($z^p$), indicating its orientation at each position. Note that the axes are smaller if they point towards the viewer or opposite (off the displayed plane). Positions (black dots) are plotted with a constant frequency, which was reduced to $\frac{1}{3}$ of the recording frequency of 100 Hz, due to clearness reasons.

On the side view plot (Fig. 5, a), $y^p$ and $z^p$ are almost drawn in full length, whereas $x^p$ is short. It can be reasoned that the pebble is oriented almost horizontally before the motion begins. By comparing the three plots of Fig. 5, one can observe that the probe is slightly tilted around the $y^p$-axis towards the left side.

During phase A, mainly vertical displacement can be seen in Fig. 5 (a). Projected on the $y^{rel}z^{rel}$-plane, the trajectory shows an almost linear pattern. On the $x^{rel}y^{rel}$-plane (fig. 5, b), a slight rightward displacement is visible. The pebble's orientation
remains more or less constant and only minor tilting can be observed as the $y^p$-axis (green) points a little downwards. At the end of phase A, the pebble rotates back again.

In phase B, a transition to a curved trajectory can be observed in Fig. 5 (a). Interestingly, a major change in movement direction emerges on the top view at diamond I as well (Fig. 5, b). Whereas the pebble moves slightly to the left during phase A, a change in movement direction towards the right is induced at this point in time. Additionally, a slow rotation of the
pebble can be observed in phase B mainly around the $y^p$-axis. This rotation contains portions around the other axes as well. Note also that from the beginning of phase A to about the middle of phase B (approx. 0.5 m on $y^{rel}$), the distance between the small black dots (indicating its position) increases, indicating increasing velocity given a constant rate of displaying the position (33.3 Hz, see above). Afterwards, the distance between the position points decreases resulting from the deceleration of the pebble.

At diamond II, a major transition was identified above for both, raw data and movement characteristics. The same transition is obvious Fig. 5 as well. While axes configurations only varied slightly during phases A and B, pronounced changes can be seen during the phases C to E. The whole complexity of the rotation in these phases can be recognised by comparing the three plots of Fig. 5. It is visible that the pebble rotates around all axes. Further, the distances between single black dots increases again, implying a repeated acceleration. Although these rotations were not completely documented by the high-speed
sequence (Video 1), the reconstructed 2D- and 3D-trajectories reveal the complex rotation that was induced by a sudden impact at diamond II (compare also Fig. 3 and Fig. 4).

Phase F is again characterised by relatively small but continuous changes in axes orientations and by an almost linear trajec-
tory pattern. The distances between the black dots decrease further, reflecting the decreasing velocity. In addition, Fig. 5 shows that the pebble's orientation at the end of motion differs significantly from its starting orientation. The pebble is strongly ro-
tated as the $x^p$- and $y^p$-axes point towards the starting position. This could not be identified in the raw data, plotted in Fig. 3. Here, only little changes in ACC readings were identifiable. This reflects critical states, where different orientations lead to





similar (or equal) axes readings. To derive the correct orientation, advanced techniques have to be used, like a quaternion-based approach (compare e. g. Hanson, 2006). Apart from that, the $z^p$-axis points – slightly tilted – in an upward direction. Note also that the flume bottom is inclined by 20 ° (compare Fig. 2). Therefore, the linear trajectory reflects parallel motion along the flume bottom.

## 4  Potentials and limitations of the Smartstone probe

### 4.1  Trajectories of multiple pebbles in one experiment

A detailed analysis of the reconstructed motion behaviour of a single clast within a moving mass was given above. Beyond that, three more pebbles were equipped with Smartstone probes in the same experiment and their trajectories could be reconstructed in the same way as for pebble 4. Consequently, the four trajectories can be plotted together within one diagram providing that a higher-ordered reference system is applied (see Sect. 2.2 and Fig. 1).

Fig. 6 shows the reconstructed spatiotemporal trajectories of four pebbles that were equipped with a probe. Note that vertical and horizontal axes of the diagram are drawn on same scale and the duration is colour-coded relative to the start of movement. The first motion of the four analysed pebbles was set to 0.0 s (pebble 1). Therefore, both time and position coordinates differ from Fig. 3 and Fig. 5. Thick grey lines give the dimensions of the flume construction including the storage box with the simplified material body prior to the start. It is visible that pebble 1 and 2 were embedded into the material (see Fig. 2). Pebbles 4 and 3 were placed at the surface of the.

Looking at the four trajectories, it can be seen that the path of pebble 3 falls remarkably steeper than the others. This results in a reconstructed depositional position that is below the flume bottom. Here, the reconstruction obviously produces erroneous results. Comparing the end of the trajectory and the true deposition of pebble 3, the overall length (projected length of the 2D displacement on $y^f$/$z^f$-plane) fits quite well to the measured one. It seems that only the inclination of the reconstructed trajectory was misinterpreted by the algorithm. A wrong estimation of the initial orientation is considered to be the main disturbance for the wrong orientation of the trajectory. A false reconstruction might occur if the probe did not record the stationary conditions prior to the start of motion. However, the time series of pebble 3 was found to be complete after a detailed review. Another reason could be that – contrary to the other clasts – pebble 3 might be strongly inclined under stationary conditions prior to the start of the experiment. This would result in a wrong estimation of the vertical direction leading to an overestimation of the vertical acceleration component and the double-integrated vertical displacement.

The other trajectories on the other hand show patterns that are reasonable compared to the reference measurements: The two embedded pebbles covered a shorter distance than the pebble placed at the surface. Additionally, at the same point in time – for instance at approx. 1.0 s since the start of the experiment (again red coloured) – pebble 4 has travelled approx. 0.4 m further than the embedded ones. These observations are consistent with basic physical laws as objects on higher position exhibit higher potential energy that can be transferred within the motion process. Moreover, the embedded pebble 1 displaced roughly 70 % of the resulting distance (approx. 0.35 m of 0.50 m) within approx. 0.5 s (light blue colours). It is conspicuous that during this phase mainly vertical displacement occurs, whereas the latter 30 % of its trajectory it moves more or less parallel to the



inclined flume plane. For this part of its trajectory the pebble needs another approx. 0.8 s until it finally deposits. It can be reasoned that a strong gradient in velocity magnitude after the transition from mainly vertical to lateral displacement occurs. This motion behaviour was only observed for pebble 1 in this experiment. Contrary to pebble 1, the trajectory of pebble 2, which was embedded as well, shows an uniform pattern. In addition, a smooth velocity gradient can be observed, indicated by gradually changing colours. Therefore, the two pebbles, which were embedded at opposite sides of the material (compare

Fig. 2), show dissimilar motion patterns. Probably the motion behaviour is depended on the pebble's distance to the opening board of the flume. Clasts that are further to it – such as pebble 2 – are surrounded by more material confining a free motion. Therefore, almost free fall conditions will be easier to achieve closer to the opening as in the case of pebble 1. This is also in agreement with the reconstructed trajectory of pebble 4 that shows a similar pattern until approx. 1 s after the start.

Contrary to the uniform trajectories of pebbles 1 and 2, saltation can be recognised approx. 1.15 s after the start between

approx. 1.3 m and 1.6 m horizontal distance ($y^f$) for pebble 4. This feature is the result of the complex rotations that were identified in motion raw data (Fig. 3) as well as the derived movement characteristics (Fig. 4) and were finally visualised in Fig. 5. Now, comparing all valid trajectories within the flume reference system, this bumping pattern of pebble 4 becomes very notable. During this phase the general lateral motion along the inclining plane – driven by gravitational acceleration and decelerated by friction – is interrupted. In fact, the pebble moves more or less horizontally before it falls again and proceeds

its 'normal' motion. This extraordinary motion pattern was initiated by a strong hit visible in raw motion data (Fig. 3, a) at diamond II. Although the trigger can be identified in the raw data, its actual meaning and, subsequently, a suitable interpretation is only possible if the movement is visualised within the correct spatiotemporal context. Hence, a rudimentary plotting of motion raw data is not sufficient to describe and interpret geomorphic movement processes adequately.

## 4.2   Probe restrictions and analytical limitations

The current Smartstone probe v2.0 exhibits one main drawback. Since the probe development focussed on the minimal possible size, only a small button cell battery can be used as energy supply. This means that battery life is restricted, especially under cold conditions. Consequently, the batteries had to be changed before the data was read out during the experimental campaign. This resulted in a pronounced battery wastage. During the future development of the Smartstone probe, alternative options for energy supply will have to be evaluated.

Beyond the battery issues, some probes seemed to be error-prone. As indicated before, it was not possible to record a calibration sequence with one probe. Although this probe was handled in the same way as all others, it was somehow damaged. In this case, an initiation of the recording mode was not possible anymore. The reason for this could not be evaluated.

Despite this, all other probes could be used during the experiments and the data could be used to reconstruct the 3D-trajectories as described above. When comparing the end of each valid trajectory and the true depositional position, a particular

deviation of several centimetres is visible. In all cases the reconstructed trajectory is shorter than the actual distance that was covered by the pebble. It can be reasoned that the displacement is generally underestimated by the calculations. This is in contrast to the analytical results of Gronz et al. (2016), where mainly the clipping of ACC readings lead to an overestimation of the displacement. In the present experiment, clipping was not observed due to the enhancement of the ACC recording range.





Another explanation for an erroneous displacement derivation might be an incorrect duration of the non-stationary period.
During data analysis, beginning and end of the non-stationary period were set manually since the primary filter approach of x-io
Technologies (2013) was not applicable for these kind of motion processes. As described before, the algorithm was originally
developed to track the human gait that is characterised by uniform and distinct motion patterns. Since the motion behaviour
of clasts within a moving granular material possesses a higher level of complexity and is less predictable, the necessary filter
parameter settings change significantly for each recorded motion. Consequently, for each pebble and in each experiment,
multiple filter parameters would have to be found. Therefore, a manual setting was considered to be more effective. Since the
start of the motion process is clearly visible in the sensor raw data (compare Fig. 3), the beginning of the non-stationary period
can be set easily. On the other hand, the motion process mostly declines gradually and slowly (compare sections 3.1 and 3.2).
Therefore, the end of motion was difficult to identify. In a consistent way, the end of non-stationary periods was set to a timestep,
were almost no rotation was recorded by the GYR. Around this time at the end of the recorded sequences, ACC readings were
low as well. This indicates that more or less only gravitational acceleration was acting on the pebble and acceleration due to
transport motion is negligible. Therefore, it is unlikely that a further transport of the pebble and, consequently, an unrecognised
displacement occurs. Accordingly, somewhat shorter or longer durations at low acceleration magnitudes do not significantly
influence the derivation of displacement. Although the manual definition of the end of motion will always be debatable, the
effect of a slightly longer or shorter motion is considered to be extremely small.

Other explanations for the deviation between true and calculated distances are (i) errors due to integration and (ii) imprecise
estimations of the probe's orientations. Besides others, these errors were discussed in detail by Gronz et al. (2016). Because
the raw data with a finite sampling rate and resolution is integrated twice in each timestep, a deviation will always occur and
will increase with both, time and covered distance.

In the present study, the deviation is considered to be mainly caused by imprecise orientation estimations. As deducted by
Gronz et al. (2016), an orientation error of only 1 ° will lead to an erogenous displacement of approx. 0.34 m after 2 s of
motion. Compared to the former experiments of Gronz et al. (2016), MAG data was not recorded and could therefore not be
included into the sensor fusion analysis. Keeping this in mind, a deviation between true and reconstructed displacement of
approx. 10 % (pebble 4, see Sect. 3.2) demonstrates a good quality of the applied methodology. Especially the avoidance of
clipping errors contributes to this promising result. This was achieved by enhancing the ACC measuring range from ± 4 g to
± 16 g (compare Gronz et al., 2016).

### 4.3   Possible ways to enhance the probe accuracy

A comparable low deviation was achieved by merging only ACC and GYR data. Therefore, it can be reasoned that a further
enhancement would be possible by the inclusion of MAG data. This would also allow to display the trajectory in a global
reference system. The effect of these enhancements will be scope of further studies.

A further accuracy enhancement of the trajectory reconstructions could be achieved by applying methods that are well-
established in different disciplines like pedestrian navigation or mobile robotics, like KALMAN-filtering or MARKOV Local-
ization. The latter approach uses a probabilistic description of the possible position of the pebble as a density field, which is



updated in the upcoming timestep(s) (Fox et al., 1999). Not only motion raw data could be used but also information about the surrounding relief (flume geometry), for instance. Additionally, information of the pebble (e. g. geometry, specific unit weight)

or the surrounding material could be implemented. Further studies will have to evaluate which of these information will lead to an even better reconstruction of the trajectory. Another aspect worth mentioning might be the automatic indication of the motion mode like proposed by Becker et al. (2015).

### 4.4  Potentials of the Smartstone probe

Although the exact depositional position could not be reconstructed quantitatively, which is particular pronounced for pebble 4,

qualitative depositional features were found correctly. For instance, pebbles 1 and 2 were embedded before and were also within the deposit after the experiment. This can be concluded from the relatively large vertical distance between the end of trajectory and the flume plane. Contrary to that, pebble 4 was originally deposed directly at the inclined plane which is also reproduced by means of probe data.

Furthermore, the complex rotational movement of pebble 4 can be identified in the reconstructed 3D-trajectory. This par-

ticular feature becomes also clear, if one compares the trajectory of pebble 4 with the other reconstructed transport paths (see above and Fig. 6). Contrary to pebble 4, the other clasts follow relatively simple trajectories. This can be explained by the position of these clasts embedded within the body. Therefore, the motion was strongly confined. Although only data of four probes could be analysed, prominent differences of the motion behaviour dependent on different position within the moving mass could be found.

These results demonstrate the potentials of using *in situ* motion sensors to characterise artificial landslide movements. Contrary to external observation methods, as high-speed videos or laser techniques (e. g. Manzella and Labiouse, 2009), the internal measurement supplies continuous movement characteristics for a single particle in 3D-space. The Smartstone probe thereby overcomes the issue of confining the motion process by wires. This problem emerged in many experimental studies that tried to measure the internal deformation or movement characteristics (e. g. Moriwaki et al., 2004; Ochiai et al., 2004, 2007; Olinde

and Johnson, 2015). Although the influence due to wired sensors seems small, its exact effect on the motion process can not be determined. By means of unwired sensors this methodological inaccuracy can be avoided.

During the last years, both wired and unwired sensors (IMU or other combinations of ACC, GYR, MAG) were used to observe geomorphic motion transport processes. Ooi et al. (2014, 2016) for instance used it to study the initiation process of small-scale laboratory landslides. They used the ACC data for qualitative interpretations concerning the timing of landslide

initiation. Additionally, they interpreted a rotational failure process from changing vectorial portions of the gravity vector on different axes. Nevertheless, a quantitative characterisation was not carried out. The potential of recording motion data of geomorphic movements is far beyond a simple plotting of raw data. In fact, it allows the sampling of movement characteristics. Therefore, the recording and analysis of geomorphic motion data expanses the toolkit of landslide science.

Recently, Spreitzer et al. (2019) used a similar approach as in the present study to derive Euler angles of moving wood

in laboratory experiences. They illustrate the suitability of this technique to characterise certain transport features. However,




a deviation of movement characteristics or a reconstruction of the trajectory was not carried out. The present study, though, demonstrates that movement characteristics are essential to describe geomorphic motion processes adequately.

These derivations are possible even if only acceleration and rotation data is recorded by means of a 6-DoF-probe. Further, a full 3D reconstruction of multiple trajectories was achieved. This allows a comparison between different parts of the moving
mass. Accordingly, the present study demonstrates that the 'sampling of motion' during geomorphic movement processes is possible.

## 5   Conclusions and final remarks

Laboratory experiments are a common tool to study landslide processes in detail. However, a critical – but also difficult – task is to capture the internal dynamics of the moving material. In the present paper, we presented the autonomous Smartstone
probe v2.0 that is able to measure *in situ* motion data of single clasts moving embedded or superficially in/on a landslide mass. The main conclusions of the present study can be summarised as follows:

– The Smartstone probe in its recent version fulfils all requirements to use it as an additional tool to capture single clast movements in laboratory-scale artificial landslides. Especially its size and measuring range satisfy the development aims. Additionally, the probe dimensions are adaptable to other experimental conditions or research objectives. The Smartstone
probe can be used under dry and wet conditions and is able to move, record and transmit data autonomously and wirelessly. The communication works under low power consumption via active RFID (contrary to high power consuming wireless LAN).

– Already the calibrated raw data offers broad insights into the motion process. By means of the acceleration and rotation time series the motion sequence of pebble 4 could be subdivided into six phases with individual motion behaviour.
A qualitative interpretation of the raw data reveals stationary-, (almost) zero-g-, translational- and rotational motion modes. Moreover, a complex rotational motion could be identified, which is initiated by strong acceleration-peaks and characterised by angular velocities.

– Using sensor fusion algorithms, the motion sequence can be quantified within a local reference system. Quantifying motion requires a calculation of the movement characteristics ($a^{rel}$, $v^{rel}$, $s^{rel}$). This could be achieved satisfactorily by
merging acceleration and rotation data. Sensor fusion allows the *in situ* measurement of movement characteristics independently from visual contact to the object of interest and without confining wires. Therefore, smart sensor technology provides the opportunity to sample movement characteristics directly within a moving mass of individual clasts.

– By means of the calculated movement characteristics, a full 3D reconstruction of the trajectory was possible. This is a great tool to visualise motion and facilitates the qualitative interpretation of transport processes.





– Finally, it was demonstrated that multiple Smartstone probes can be applied in one experiment. To take the metaphor, this allows to take multiple motion samples from different parts of a moving landslide body. This opportunity may shed light on the internal dynamics and potential deformation of moving landslide bodies.

Although the Smartstone probe prototype has to be further improved (see Sect. 4.2), the present study indicates a methodological enhancement by means of smart sensors and sensor fusion algorithms. Further studies will have to focus on the

comparability to other well-established methods, for instance PIV.

Beyond that, new analytical solutions have to be found to deal with motion data in geoscience. Therefore, future studies will focus on the question how motion characteristics like the transport mode can be classified by means of these kind of data.

*Video supplement.* A supplementary high-speed video (Video 1) is available online.

*Author contributions.* B. Dost planned and carried out the present study. He prepared and conducted the experiments, analysed, visualised

and interpreted the data and wrote the present article. O. Gronz developed key parts of the experimental setup, helped to conduct the experiments, created the high-speed video, interpreted the data and wrote parts of the article. M. Casper was involved in the development process of the Smartstone probe and the experimental setup, supervised the work, interpreted the data and reviewed the article. A. Krein helped to develop the experimental setup, interpreted the data and reviewed the article.

*Competing interests.* The author declare that no competing interests are present.

*Acknowledgements.* The authors thank the Luxembourg Institute of Science and Technology for partly funding the experimental setup in the framework of the internship of Bastian Dost and the Smartstone probe v2.0. Many thanks to Yannick Hausener for his great work constructing the experimental flume. The authors additionally thank the former students Julius Weimper, Björn Klaes and Christoph Löber for their help conducting the experiments. Sieving analysis was kindly carried out by Grundbaulabor Trier Ingenieurgesellschaft mbH, Germany.



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



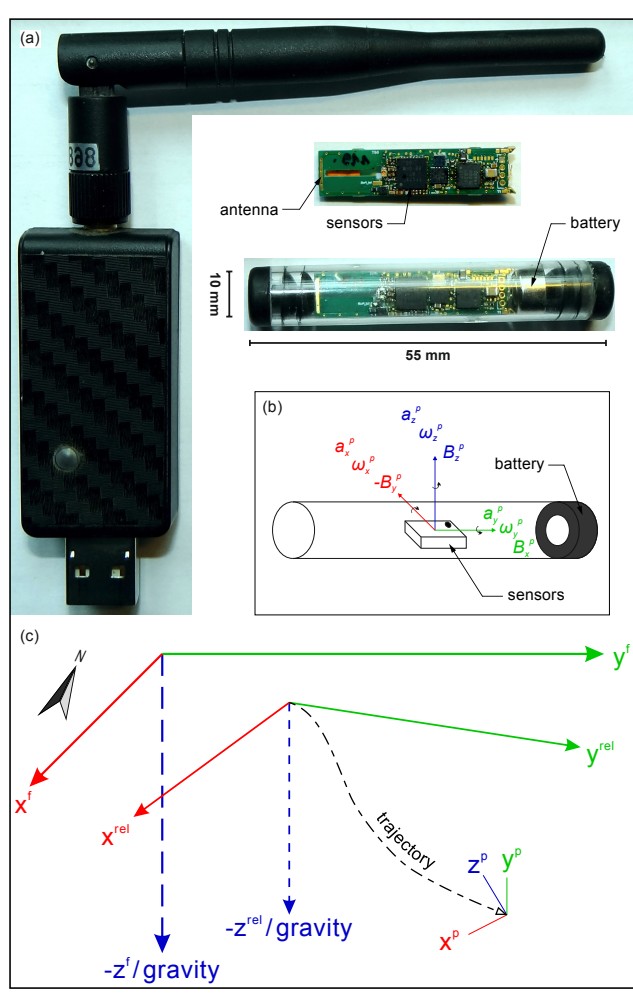

**Figure 1.** Smartstone v2.0 hardware kit and technical conventions. (a) Hardware components including an USB-gateway with antenna for communication between a computer and the probe. Electronic components within a plastic tube, hosting the triaxial sensors (compare Table 1) and the internal antenna. (b) Axes conventions of the Smartstone probe v2.0. (c) Reference systems as used in the present study.

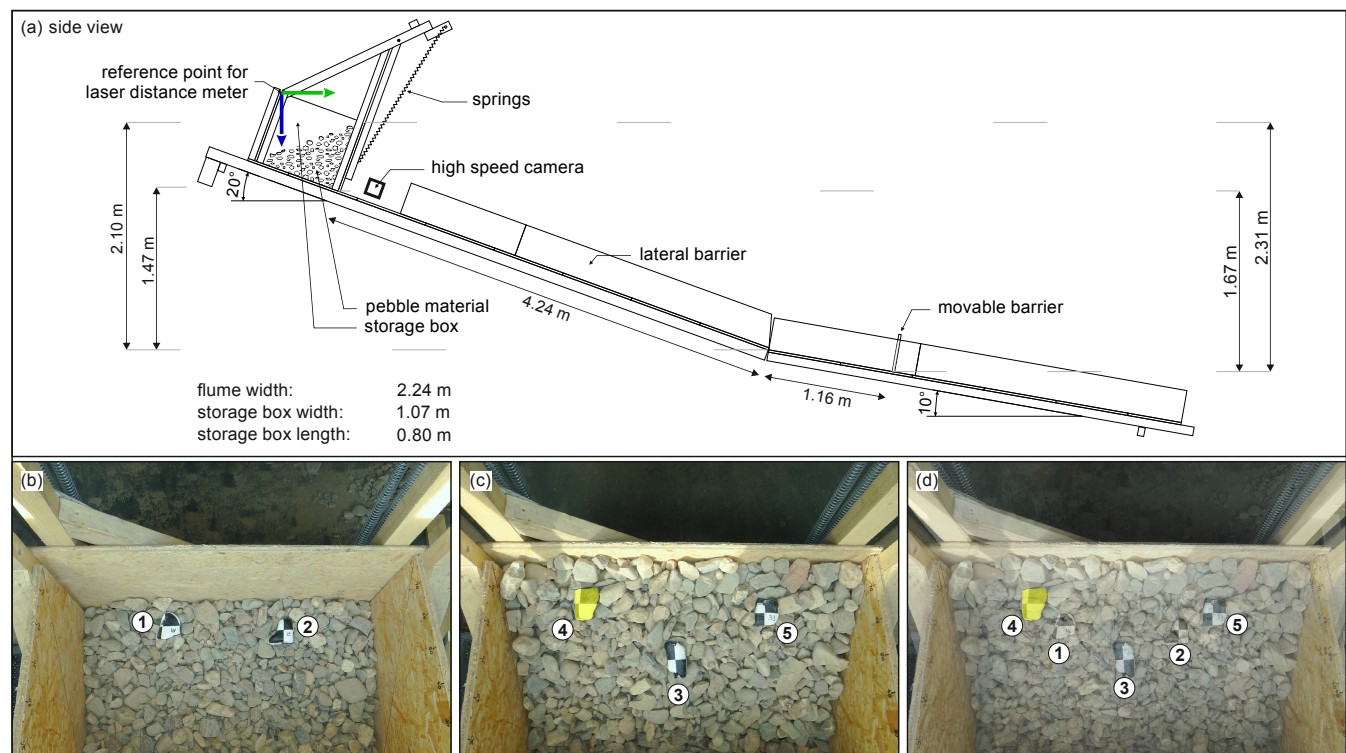

**Figure 2.** Experimental Setup. (a) Simplified sketch of the laboratory flume. Green and blue arrows mark axes of the flume reference system in $y^f$- and $z^f$-direction, respectively. (b-d) Starting positions of the five equipped embedded (b) and superficial (c) pebbles. (d) Overlay of (b) and (c) showing the initial positions of all equipped pebbles prior to the experiment. Pebble 4, whose data is plotted in Fig. 3 to 5, is highlighted in light yellow in (b) and (d). Note that pebble 5 could not be included into the analyses due to its damage.

**Figure 3.** Calibrated raw data of pebble 4. Data is plotted versus relative time since the start of motion. Stationary periods are indicated by white bars, motion by a black bar at the top of each plot, respectively. Vertical bars in light yellow, grey and red mark particular phases (A-F) within the motion sequence (for description see text). Numbered diamonds indicate distinct points in time (see also Fig. 4). (a) ACC data, (b) resultant acceleration magnitude, (c) GYR data. Curves in (a) and (c) show recordings for each probe axis, respectively.

**Figure 4.** Movement characteristics and high-speed captures. (a-c) Time series of the derived movement characteristics for (a) translational acceleration, (b) translational velocity and (c) translational displacement. The three curves in each plot give calculated time-dependent values for each axis of the relative reference system (as defined in Sect. 2.2). Motion phases and distinct time points are indicated as in Fig. 3. In (c) green and blue dots indicate the true displacement components in $y^f$- and $z^f$-directions measured by means of a laser distance meter (for explanation see Sect. 2.2). (d) High-speed captures at time points diamond I and II. The data of (a-c) was recorded with pebble 4, which is highlighted in light yellow and labelled in (d).
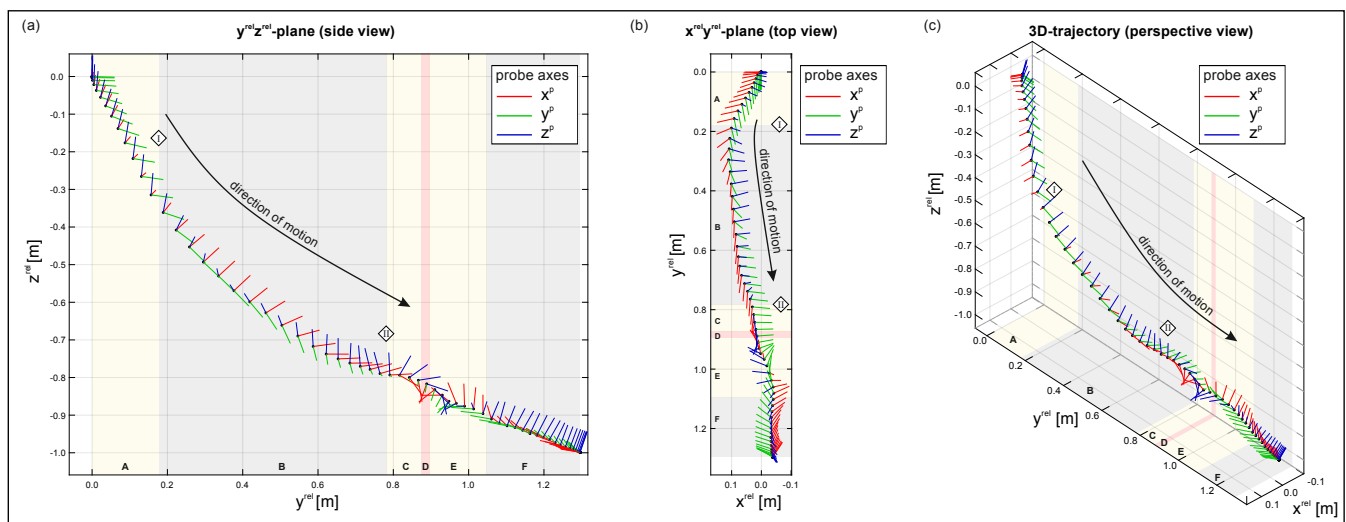

**Figure 5.** Visualisation of the reconstructed trajectory within the relative reference system. Red, green and blue coloured lines indicate probe axes and display its orientation within the relative reference system as defined in Sect. 2.2. Motion phases and distinct time points are indicated as in 3. (a) 2D-side view ($y^{rel}z^{rel}$-plane), (b) 2D-top view ($x^{rel}y^{rel}$-plane), (c) 3D perspective view.



**Figure 6.** Reconstructed spatiotemporal trajectories and true depositional positions of four pebbles within the local flume reference system. 2D-side view ($y^{rel}z^{rel}$-plane). In addition, the idealised flume construction and the pebble material body prior to the experiment is drawn in light grey colours. Note that the equipped pebbles are not drawn in scale due to clearness reasons. The trajectories are colour coded, where the colour represents the relative time since the start of the experiment (first motion, pebble 1).



**Table 1.** Technical specifications of the Smartstone probe v2.0 as provided by Bosch Sensortec GmbH (2014, 2015) and manufacturer information from Smart Solutions Technology GbR.

| Component | Sensor | Measuring range | Noise | Sample-rate |
|-----------|--------|-----------------|-------|-------------|
| Bosch BMI 160 | ACC | $\pm$ 16 g | 1 mg | adjustable: |
| | GYR | $\pm$ 2000 $^{\circ}$ $^{-1}$ | 0.04 $^{\circ}$ $^{-1}$ | 12.5 Hz, 25 Hz, 100 Hz |
| Bosch BMC 150 | MAG | $\pm$ 1300 $\mu$T (x-,y-axis) | 1 - 2 $\mu$T | |
| | | $\pm$ 2500 $\mu$T (z-axis) | 1 - 2 $\mu$T | |


**Table 2.** Motion phases of pebble 4 as displayed in Fig. 3-5 and high-speed video. IDX gives the index of data samples and $t_{Start}$ gives time in seconds since the start of motion. Frame indicated the frame number as displayed in Video 1. For other columns see description within the text.

| Phase | IDX | $t_{start}$ | Frame | $s_y^{rel}$ | $s_z^{rel}$ | $s_y^{f}$ | $s_y^{f}$ |
|-------|-----|-------------|-------|-------------|-------------|-----------|-----------|
| | [-] | [s] | [-] | [m] | [m] | [m] | [m] |
| A | 12 | 0.000 | 64 | 0.000 | 0.000 | - | - |
| B | 51 | 0.389 | 259 | 0.1774 | -0.3464 | - | - |
| C | 102 | 0.898 | 513 | 0.7823 | -0.7955 | - | - |
| D | 113 | 1.008 | 568 | 0.8739 | -0.8094 | - | - |
| E | 116 | 1.038 | 583 | 0.8958 | -0.8217 | - | - |
| F | 143 | 1.307 | 718 | 1.0957 | -0.9326 | - | - |
| end | 195 | 1.826 | 977 | 1.2962 | -0.9988 | 1.4240 | -1.1090 |