# Peer review of "The Potential of Smartstone Probes in Landslide Experiments: How to Read Motion Data"

_Natural Hazards and Earth System Sciences, 2020_

## Referee Comment (RC1) · Anonymous Referee #1 · 26 Mar 2020

Please find a review of the manuscript titled "The Potential of Smartstone Probes in Landslide Experiments: How to Read Motion Data", submitted by Dost et al. to NHESS. The authors present the use of "Smartstones", pebbles equipped with motion probe, to characterize motion features of single clasts within artificial laboratory-scale landslides. The authors well present the literature of landslides and specifically in the lab, and they identified the missing part, of describing the motions modes of single clasts during the landslides. I think that they present a promising branch of the research of landslides, thus, I recommend on accepting the paper, but I also have some major comments and thus recommend on major revisions, and then consider acceptance.

As an NHESS paper, I would expect a more rigorous explanation of the experimental scaling to landslides. I understand that the main goal is to proof the concept of the

[Figure]

Smartstones method, however it is placed in a lab simulation of landslide. Moreover, the paper is structured as a laboratory experiment of landslide, including the reasoning in the introduction, and a very detailed description of the transport of the gravels, and the different modes of their motion. Thus I think that the reasoning of the specific experimental setup should be explained. Also, upscaling considerations to the larger scale, real world, landslides should be discussed including scaling analysis, as is expected from physical experiments.

I suggest to send the manuscript to grammatical editing. Also some typos should be cleaned from the manuscript.

Repetitions between figure captions and main text – unneeded redundancy.

Add a bit more about the implication of this method in the Abstract

L11 – mention the size\type of pebbles is more interesting then the entire mass.

L79 – what is the former version? – maybe I missed."

L124 – needed?

L105 - The sampling rate (100 Hz) is low. Most probably, it cannot record the sharp impulse during a collision with another rigid body. Miss recording such impulse can lead to significant errors in the velocity and position, calculated by integrating the recorded acceleration. The authors should address this issue, provide an estimate of a typical collision duration in their experiment, and show that it is longer than 1/sampling rate. In case the above condition is not fulfilled in the experiment, the authors should explain the implications. A short discussion on the sampling rate in a broader context of a real landslide can be illuminating for the reader.

L133 - It appears that the system of coordinate of the ACC is not following the convention of the "Right-hand rule". Can the authors comment on that. In any case this is an important information for the reader.

L137 - 1. The term "higher-order" may not be the best choice as the position of a body/point is always relative. 2. The "real" axis system is not suitable for comparison of different probes; the "flume" system suit this purpose; this is why the authors used it for the graph in Figure 6 that compares the movements of different probes.

L223 – Can you clarify 0.0 g means?

L225 – The whole section (3.1) is very long and tedious. The formulas are trivial, in any case, the authors do not use the projections of g in the discussion.

L432 –the end of the sentence is missing.

L437-442 - The most probable reason for the wrong trajectory of pebble 3 is miss recording of collision with another pebble due to the slow sampling rate of the used IMU.

L446-447 This statement does not fit the description of the behavior of one body, out of many, in a multi-body system where collisions between bodies redistribute the energy of the system in a random way.

L472-473 – Too much details.

L510 - In the present study, the probe monitored movements over a short period of~ 2 sec. A brief discussion regarding the expected error in retrieving the trajectory over more extended periods can help to assess the type and scale of landslides that can be monitored in this way.

Figure 3 – change the axis title to the same side.

Figure 5a,c – add the flume reference as well.

Fig 6 - color coding has a few cycles so it is not injective and a bit hard to follow, maybe add time stamps at the end of each cycle?

2020-61, 2020.

---

## Referee Comment (RC2) · Anonymous Referee #2 · 29 Mar 2020

The manuscript entitled "The potential of Smartstone probes in landslide experiments: how to read motion data" reports data and results from a set of laboratory experiments, in which the movement of selected individual tracer stones within a landslide was recorded by means of a "Smartstone probe". The presented technology is interesting, and the manuscript shows the potential of this technology for analyzing the movement of individual grains within a granular flow. However, the manuscript focuses solely on the interpretation of the recordings and one of the main findings is that the technology requires further improvements. For example, the manuscript indicates that the system is not really stable (one sensor out of five did not work appropriately, and another produced false results). Moreover, the statements at L483 and following indicate that the data analyses need to be further improved and that the current analyses are, to

some extent, premature. What is lacking, in my opinion, is how these data can be used to further develop theories regarding granular flows (the latter have not been reviewed in the manuscript). In this context, it is good to see that the technology advances and that it is shown how digitalization can be helpful to tackle complex problems, but the data needs to be analyzed accordingly so that the manuscript could be published as a research article. I would also like to note that, although the manuscript has been prepared with great care, the manuscript needs to be improved language-wise and stylistically, so the that the contents becomes clearer. This becomes obvious from my large amount of detailed comments below. To summarize - the presented material is interesting, but I do not see substantial and original scientific results which warrant publication of the manuscript as a research article. In contrast, the manuscript "reports new developments, significant advances, and novel aspects of experimental and theoretical methods and techniques which are relevant for scientific investigations within the journal scope". This statement has directly been copied from the NHESS-website defining the manuscript type "Brief communication". These aspects are covered by the manuscript and I therefore recommend that the authors shorten the manuscript significantly and resubmit it as a brief communication.

Detailed comments - many of my comments highlight issues regarding the presentation of the material and the language - therefore not all of my comments are of major nature. Note also that I wrote the comments before finally recommending resubmission as a brief communication. Nonetheless, I decided to provide substantiate my above evaluation.

P1, L2: I am not convinced that every reader understands what is meant by external and internal information.

P1, L4: The first "internal" can be deleted.

P1, L6: "artificial laboratory-scale landslide" - artificial may be deleted

P1, L7-10: Is this detailed information really adequate for the abstract?

P1, L11: I partly disagree with this statement - the movement of individual pebbles was observed with the Smartstone probe, but not the motion of 520 kg of the pebble material...

P1, L12: Which mass is meant - the mass of the pebble-material?

P1, L13-21: This is mainly a description of what has been done - what is lacking is a more generalized description of the results - i.e. the novelty aspect of the study should be better highlighted.

P2, L28: In my opinion the paper would benefit from additional considerations (including a review) on granular flow mechanics and how these can be described using the sensor data.

P2, L32: What is meant by "some depositional features"? A more general description of the landslide processes would be helpful.

P2, L36: What is meant by precess? Is "process" meant? I am also not sure that I understand what is meant here.

P2, L37: Include year for reference Okura et al.

P2, L38: My understanding of the word "collusion" seems to be different from the understanding of the authors. Maybe "collisions" is meant by the authors?

P2, L40: The authors use specific terminology which has not been defined before. As indicated above, a more general description of landslide mechanics and granular flow would be helpful (also for the better understanding of the subsequent passages).

P2, L48: Please specify what kind of 2D section of the body is meant.

P2, L51: This is true - but is the information also relevant for the description of the movement of the granular material?

P2, L57: Replace "got" by "became"

P2, L58: What is the technical aspect of so called 'smart tracers' for natural transport? These tracers can be helpful to collect data for the description of natural transport processes but have no effect on natural transport... (please try to be specific language wise throughout the manuscript).

P3, L59: What is the significance of this sentence?

P3, L65 and following: This could be more concise.

P3, L75: As already mentioned - the physics of the movement of granular material should be better highlighted. I fully agree with the next statement that the manuscript focuses on sensor application (i.e. on the method), and this is exactly why I see limits regarding the significance and novelty of the scientific findings. Therefore I finally recommended to resubmit the manuscript as a brief communication instead of a research paper.

P3, L79: Please improve the description of the objectives.

P4, L93: Why "mainly"?

P4, L100: Why "available"?

P4, L122: Different dimensions have been mentioned before (L96) which is confusing - I find it also confusing that the dimensions are given only "approximately" - what are the exact dimensions?

P4, L124: What is meant by "lager objects"? Is "larger objects" meant? Please check the language throughout the manuscript (I stop here giving comments on the language).

P5, L125: Sentence starting at L124 - what is the significance of this sentence for the study?

P5, L127: This information could be given in the Acknowledgements or incorporated in the above text. The sentence that further improvements are planned already indicates

that the presented findings are premature (similar statements are given at the end of the manuscript).

P5, L137: The reference systems could be introduced better... I find the presentation of Figure 1c rather confusing (as the reference system 'f' is only introduced at L150).

P5, L140: Why is 'relative' in italics? I don't think that "donated" is adequate terminology... (throughout the paper)

P5, L143: Why 'probably'?

P5, L151: I don't understand - please improve the description and be more precise (e.g., it is only mentioned below that z follows the direction of gravity).

P5, L153: What is meant by higher-order global system (i.e., why higher-order)?

P5, L154: Why was this not done?

P6, L162: I am not sure that I understand what is meant by (i) as well as (ii and iii).

P6, L163: 'describes' must be 'describe'.

P6, L165: All this remains a black box and could be explained in some more detail. I also find that the 'damaged probe' is mentioned too often...

P6, L170: Some more details would be desirable (or at least references where relevant information can be found). I find this important section rather short.

P6, L178: What is meant by 'various studies'?

P6, L180: A more precise length is given in the figure, which I find confusing (why approx. when the exact length is known?).

P6, L181: I find this difficult to read - please improve language.

P7, L206: Please check language. What was the accuracy of these measurements?

P7, L210: This statement defines the scope of the study - the demonstration of the

applicability of the sensor to monitor the motion of individual pebbles within a landslide - this calls again for publication as a brief communication instead of a research article.

P7, L211: The information on the date should be presented in Section 2.4 (is this information important)?

P7, L216: As mentioned above, the calibration of the sensor remains partly foggy, and I am not sure that the terminology "calibrated raw data" is adequate - it seems to me that raw data are presented which were obtained by a calibrated sensor?

P7, L217: Could this be measured without activated IMU? If not, the statement in brackets could be deleted.

P7, L217/218: I had first difficulties to understand this sentence when looking at the figure - this could be formulated more clearly.

P8, L222: Why does the time series begin at a 'negative time'? When exactly was the landslide triggered? This should be mentioned/explained more clearly. Also, the statements relate to accelerations along the considered axes, this should be better highlighted in the text (xp and yp show low values - but strictly speaking, these are coordinate axes; the same applies to the statements given in the next sentences).

P8, L223: Why 'seems'? The data show this higher level.

P8, L230: Please improve - this is difficult to read and understand (suggestion: alpha, beta and gamma define the angle between xp, yp and zp and the gravity vector).

P8, L230 - L312: This qualitative presentation of the raw data is difficult to read and understand - it should, in my opinion, be coupled with the derived trajectory.

P8, L231: 'downwards direction' is, in my opinion, a rather confusing terminology.

P8, L234: If the pebble is stationary, it is clear that it cannot rotate...

P8, L234: I guess the landslide was triggered at 0.00 s? This should be mentioned in

the text. Define 'the change' in the plots more clearly.

P8. L235: The text implies that the zp-axis drops, which is not the case (and which is rather confusing). What drops is the acceleration value, and this needs to be described more clearly (also in the following passages).

P8, L238: This could be explained better and the motion characteristics 'free fall' and 'hampered free fall' should be defined more clearly.

P9, L248: This seems to be rather speculative - a statement like 'a conclusion might be...' is very vague...

P9, L251: Note that the values range between 1 g and -1 g (and do not correspond to 1 g and -1 g as presently stated).

P9, L257: However, I can see other spikes which are defined by two data points.

P9, L260: Indicate which axis is analyzed here.

P10, L289: 'Of cause'?

P10, L290: Why?

P10, L290: This is trivial - I would be surprised if the pebble would not change its orientation.

P10, L313: The above qualitative statements regarding the motion mode should be coupled with the considerations of the movement with respect to position and time.

P10, L315: This could be mentioned earlier.

P11, L316: Are these relationships really that simple?

P11, L317: Check language ('calculated after...').

P11, L318: Why were the angles 'received'? How can the accelerations be 'rearranged'? Please check language throughout.

P11, L320: This is a description of a method which is described in detail in the literature - it could therefore be included into Section 2.

P11, L328: 'srel' is not integrated. It is obtained from integration.

P11, 336: I am not sure that I understand what is meant here. Please improve.

P11, L337 and following: All this could be combined with the qualitative discussion.

P11, L338: I disagree with this statement: a_zrel does not increase but decrease from 0 to -9.4. It also does not do this continuously decrease as clearly shown by the fluctuations in the plot.

P11, L339: Why? This should be explained better. I also cannot see in the Figure that the pebble 'swims'. Please improve.

P11, L342: What is the significance of the maximum velocity v_yrel? Why is v_zrel decreasing afterwards while it further increased simultaneously? This is confusing.

P11, L343: Why 'covered'?

P11, L345: I disagree again - the plots for the y-axis do not really exhibit a major change. I am also not sure about the relevance of the capture of the high-speed sequence.

P11, L346: Is the 'acceleration curve' really smooth before 0.389 s?

P12, L349: This is difficult to understand and should be explained better.

P12, L351: What is the significance of this observed maximum?

P12, L353: Please explain better - this is hard to see from Figure c.

P12, L355: Figure 4 d shows photographs and not displacement plots?

P12, L355: I do not see these staircases (if plot c was meant).

P12, L372: This was mentioned before.

P13, L390: I am not sure that I understand what is meant.

P13, L390 and following: This could be combined with the previous analyses.

P14, L432: This sentence is incomplete.

P14, L437: This indicates that further work is required.

P15, L470: This is a drawback for field applications, but not really for laboratory experiments.

---

## Author Comment (AC1) · 6 Jun 2020

**Response to referee #1**

We thank the reviewer for the time taken to review our manuscript and for the useful comments. We respond to each of the individual review comments below:

Referee text is black.

Response text is blue.

Changes are red.

General comments:

I think that the reasoning of the specific experimental setup should be explained.

The first sentence of 2.4 Experimental setup has been changed to:

The experimental setup was designed regarding the following requirements: (i) an exact and rapid triggering mechanism, (ii) multiple repetitions with identical boundary conditions due to homogeneous and dry material and (iii) flexibility for future studies.

Upscaling considerations to the larger scale, real world, landslides should be discussed including scaling analysis, as is expected from physical experiments.

Scaling of laboratory experiments is a complex topic. Various corresponding articles exist for different research objectives (e.g. hydraulic experiments: Heller, 2011; landslide and debris flow experiments: Iverson, 2015; gravity-scaled and unscaled debris flow experiments: Turnbull et al., 2015; granular slide experiments: Kesseler et al., 2020). We agree that estimating scaling effects and treating them appropriately is essential to transfer experimental findings to real world scenarios. In the present study, however, these aspects were not included, because the overall applicability of the probe for experimental application was investigated, no physical properties. These aspects will be the research objective of future studies with the Smartstone.

The following paragraphs will be added as new subsection to the discussion between Sect. 4.2 and Sect. 4.3:

Scaling

A scaling of the recording ranges will be necessary if the Smartstone method is adapted to other experimental scales or velocities. Additionally, the scaling of temporal persistence of movements has to be respected as the Nyquist frequency to observe the motion without undersampling changes with the rate of movement changes (Yang et al, 2009). This means that a small pebble in a fast-moving landslide will show more abrupt changes in its velocity, trajectory and mode than a large block in a slow landslide. Thus, the ranges of the sensors and the sampling frequency have to be adjusted depending on the landslide velocity *and* the particle size. Several aspects concerning the sensor recording range for different experimental applications have to be considered:

Acceleration range: The expected acceleration depends on the velocity of the landslide, as the strongest peaks occur during nonelastic collisions of moving particles with stationary boundaries, e.g. bedrock. Thus, the range needs to be increased (by choosing a different accelerometer chip in the Smartstone) with velocity. However, to choose a gratuitously large range to avoid clipping is counterproductive, as the quantisation error will also increase, as there is only a limited number of steps within the range. A deliberated balance needs to be chosen, e.g. by performing preliminary tests.

Gyroscope range: The rotational velocity depends on the movement of the landslide but also on the size of particles. For instance, if the mass moves with 1 m s$^{-1}$, a single rolling pebble with 30 mm diameter will show a rotational velocity of 3820 °s$^{-1}$ (pebble circumference 942 mm, thus 10.6 rotations per second). Thus, the expected range can be calculated using the shortest circumference of the Smartstone's host particle and the expected landslide velocity. Again, choosing a gratuitously large range to avoid clipping will increase the quantisation error.

I suggest to send the manuscript to grammatical editing.

We will engage English copy-editing services of Copernicus.

Also some typos should be cleaned from the manuscript.

We corrected the typos mentioned by both referees. See also previous comment.

Repetitions between figure captions and main text – unneeded redundancy.

Some readers may only read captions; others will read the text and have afterwards a look at the figures. We belief, the explanation of figures should deal with both cases. Probably most readers will not do both.

Add a bit more about the implication of this method in the Abstract.

The implication of recording motion data of clasts embedded in moving artificial landslides was clarified in the abstract:

Compared to other observation methods Smartstone probes allow for the quantification of internal movement characteristics and, consequently, a motion sampling in landslide experiments.

Specific comments:

L11 - mention the size\type of pebbles is more interesting than the entire mass.

We added the size and type:

Using the Smartstone probe, the motion of single clasts (gravel size, $d_{50}$ of 42 mm) within approx. 520 kg of a uniformly graded pebble material was observed in a laboratory experiment.

L79 - what is the former version? – maybe I missed.„

The former prototype version is now mentioned earlier in the text. The reference is Gronz et al. (2016).

L124 - needed?

This information demonstrates that the Smartstone probe can be adapted to different research objectives.

L105 - The sampling rate (100 Hz) is low. Most probably, it cannot record the sharp impulse during a collision with another rigid body. Miss recording such impulse can lead to significant errors in the velocity and position, calculated by integrating the recorded acceleration. The authors should address this issue, provide an estimate of a typical collision duration in their experiment, and show that it is longer than 1/sampling rate. In case the above condition is not fulfilled in the experiment, the authors should explain the implications. A short discussion on the sampling rate in a broader context of a real landslide can be illuminating for the reader.

During the data analysis we did not notice any issues resulting from the sampling rate. Generally, the needed frequency does not depend on the duration of accelerations but on the uniformity of the movement. The sensor does not miss single strong acceleration peaks shorter than the sampling frequency due to its principle of operation: Inside the chip is a small movable piece shaped like a comb (for each axis). It is positioned inside a stationary second comb. Lateral movement due to accelerations changes the distance between the two combs, resulting in an electric capacity change, which is measured to derive acceleration. If a short, abrupt acceleration occurs, the movable comb will be displaced in any case, even if the acceleration does not last as long as the sampling period, due to its inertia. The chip integrates inherently during the sampling period. However, if the sensor is moved several times forward and backwards within a sampling period, the sensor will miss this information, as the movement's frequency is higher than the sampling frequency. Thus, the correct movement cannot be observed. Again, preliminary tests facilitate choosing the correct sampling frequency.

L133 - It appears that the system of coordinate of the ACC is not following the convention of the "Right-hand rule". Can the authors comment on that. In any case this is an important information for the reader.

Thanks for spotting the error in the figure. The figure has been modified. Of course, the axes follow the convention. Fig. 1 (b) has been modified:

[Figure]

L137 - 1. The term "higher-order" may not be the best choice as the position of a body/point is always relative. 2. The "real" axis system is not suitable for comparison of different probes; the "flume" system suit this purpose; this is why the authors used it for the graph in Figure 6 that compares the movements of different probes.

We modified the description in 2.2 beginning at L135 including the previous comment as follows:

Therefore, its x- and y-axis are also rotated. Following the right-hand rule, positive rotational directions are indicated by small curved black arrows.

To compare relative movement characteristics like distance or velocity of different probes, the inner data / coordinate system $p$ must be transformed into an outer reference system $rel$ (Fig. 1, c). The simplest way to do this is the construction of a reference system using the probe's starting position as coordinate origin. The system is defined by the sensor's inner coordinate system of the first timestep rotated so that z-axis follows gravity. The axes of this relative outer coordinate system are donated with $x^{rel}$, $y^{rel}$ and $z^{rel}$. After the motion has started, the probe's inner orientation will change while the outer reference system keeps its axes configuration. Consequently, within this reference system, it is possible to calculate the probe's orientation and the covered distance in each timestep. In Fig. 1 (c) for instance, the probe has changed its orientation significantly compared to its starting position while moving along the assumed trajectory.

However, the different probe-specific outer coordinate systems must be transformed into the same local reference system to compare different probes' trajectories. For the present study, this local

reference system is oriented towards the experimental flume (see section 2.4). Following the former conventions, the axes were donated as $x^f$, $y^f$ and $z^f$. Note that the axes orientations of the outer (rel) and the local reference system (f) may not be identical, except of $z^{rel}$ and $z^f$, as they follow gravity.

In different applications, where a global positioning is required, reference points of the outer coordinate systems must be known in the global system to determine the absolute probe position in the global system.

L223 - Can you clarify 0.0 g means?

Within the gravitational field of the earth, zero g can only be achieved if and only if an object moves in pure free fall without aerodynamic drag. In this case, the resultant acceleration vector of motion and the acceleration vector due to earth's gravitational acceleration exhibit the same magnitude. These vectors, however, point in the opposite direction compensating each other. This results in a measured acceleration magnitude of 0.0 g at a measuring device, such as the Smartstone probe.

L225 - The whole section (3.1) is very long and tedious. The formulas are trivial, in any case, the authors do not use the projections of g in the discussion.

We expect different subsets of so-called trivial knowledge at different readers. Thus, this section is an introduction on how this kind of data has to be read and interpreted. In discussions with colleagues it emerged that it is often not clear what is displayed within the graphs (e.g. Fig. 3). To our knowledge, there is no publication that describes spatiotemporal motion data adequately in a geoscientific context. Therefore, we decided to give a detailed description in this article.

L432 - the end of the sentence is missing.

Corrected:

It is visible that pebble 1 and 2 were embedded into the material, whereas pebble 3 and 4 were placed at the surface of the material (see Fig. 2).

L437-442 - The most probable reason for the wrong trajectory of pebble 3 is miss recording of collision with another pebble due to the slow sampling rate of the used IMU.

As explained above (comment L105), missing an acceleration peak is not possible due to the design of the acceleration sensor (as long as the motion frequency does not exceed the sampling frequency, which is given in this case). Therefore, this cannot be an explanation for the erroneous inclination of the trajectory. In addition, an overestimation of the vertical trajectory component occurs right from the beginning of the motion. The peak would have to be occurred before the motion starts, which obviously is not possible (not least because pebble 3 was placed at the surface of the material).

L446-447 This statement does not fit the description of the behavior of one body, out of many, in a multi-body system where collisions between bodies redistribute the energy of the system in a random way.

This is correct and we had to rethink this explanation. New text:

Okura et al. (2000) observed that blocks positioned at the front were also placed in the frontal deposition zone. In our experiment, the top pebbles travelled the longest distance. Regarding the high-speed video, the explanation is given by the tilted gate: The pebbles positioned on the top start their movement both downwards and to the right (from the camera perspective), thus not transferring energy to material formerly placed underneath in the storage box. The higher the pebbles are placed, the bigger is their overhang, resulting in less material vertically underneath. Compared to a vertical gate in Okura et al. (2000), less energy dissipation occurs.

L472-473 - Too much details.

Deleted.

L510 - In the present study, the probe monitored movements over a short period of ~ 2 sec. A brief discussion regarding the expected error in retrieving the trajectory over more extended periods can help to assess the type and scale of landslides that can be monitored in this way.

We agree that the duration of recorded motion is a critical aspect of this technique. Because the estimation of the displacement components requires double-integration, absolute errors will increase with time – depending on the motion. Therefore, longer durations will generally result in less accurate absolute results. The relative error will remain stable for certain kinds of movements. However, a general extrapolation of the expected error is not trivial, as it depends on the ratio of noise, quantisation error etc. to the magnitude of the true accelerations.

Figures:

Figure 3 - change the axis title to the same side.

Modified:

[Figure]

Figure 5a,c - add the flume reference as well.

To improve the orientation within the figure, the flume bottom (grey dashed line) has been added to Fig. 5 (a). Adding the flume bottom in Fig. 5 (c) as well would reduce the clearness of the trajectory visualisation. Fig. 5 (a) was modified as follows:

[Figure]

Fig 6 - color coding has a few cycles so it is not injective and a bit hard to follow, maybe add time stamps at the end of each cycle?

Repetition of cycles allows for a more precise identification of time compared to only one cycle. As the trajectory is continuous in time, the colour coding might not be injective, but it is distinct: The first time, red occurs again along the trajectory, must equal 1 s. However, we can add timestamps although we think that this is content overload.

[Figure]

Additional References:

Heller, V. (2011). Scale effects in physical hydraulic engineering models. In: Journal of Hydraulic Research 49 (3), pp. 293–306. DOI: 10.1080/00221686.2011.578914.

Iverson, R. M. (2015). Scaling and design of landslide and debris-flow experiments. In: Geomorphology 244, pp. 9–20. DOI: 10.1016/j.geomorph.2015.02.033.

Kesseler, M.; Heller, V.; Turnbull, B. (2020). Grain Reynolds Number Scale Effects in Dry Granular Slides. In: Journal of Geophysical Research: Earth Surface 125 (1). DOI: 10.1029/2019JF005347.

Yang, W. Y.; Chang, T. G.; Somg, I. H.; Cho, Y. S.; Heo, J.; Jeon, W. G.; Lee, J. W.; Kim, J. K. (2009). Signals and Systems with MATLAB. Springer-Verlag Berlin Heidelberg: Berlin, Heidelberg.

Turnbull, B.; Bowman, E. T.; McElwaine, J. N. (2015). Debris flows: Experiments and modelling. In: Comptes Rendus Physique 16 (1), pp. 86–96. DOI: 10.1016/j.crhy.2014.11.006.

---

## Author Comment (AC2) · 6 Jun 2020

**Response to referee #2**

We thank the reviewer for the time taken to review our manuscript and for the useful comments. We respond to each of the individual review comments below:

Referee text is black.

Response text is blue.

Changes are red.

General comments:

The presented technology is interesting, and the manuscript shows the potential of this technology for analyzing the movement of individual grains within a granular flow. However, the manuscript focuses solely on the interpretation of the recordings and one of the main findings is that the technology requires further improvements. For example, the manuscript indicates that the system is not really stable (one sensor out of five did not work appropriately, and another produced false results).

The Smartstone probe is a prototype and not ready for serial production. Nevertheless, it already has practical applications (e.g. helicopter rotor blades, building stability surveillance). Of course, for other applications like landslide assessments, some issues have to be addressed by further improvements. During the experimental campaign, a total number of 10 experiments was performed. In each run, 5 Smartstone probes were used in the same way as described in the manuscript, meaning a total number of 50 data sets. From these, a total number of 36 data sets were successfully recorded and could be analysed. This means a rate of more than 70 %, which we belief is quite good for a prototype. The critical task is not to build the devices in a better way. Nevertheless, a deeper understanding of the data is necessary. This is discussed in detail in the text.

Moreover, the statements at L483 and following indicate that the data analyses need to be further improved and that the current analyses are, to some extent, premature.

We agree, as mentioned in the discussion. But from our point of view, also single (complete) steps in the knowledge gaining process should be published (see next comment). To our knowledge, this is the first article that tries to go beyond only plotting the acceleration time series (which has been done quite often before) and describing trivial statistical properties of the time series. We describe how to interpret the time series from a motion's point of view.

The presented material is interesting, but I do not see substantial and original scientific results which warrant publication of the manuscript as a research article.

We disagree with this statement. Scientific research progresses step by step. We belief it is good scientific practice to publish single step progress in detail. In addition, we belief that the presented method, corresponding results and analysis are scientifically relevant due to the following reasons: (i) We identified an important methodological gap in studying landslide motion processes. Only a sensor-based autonomous instrumentation allows for an observation of the interior of a moving landslide mass, as outlined already in the introduction. (ii) We documented a significant methodological improvement. The recent probe version was technically enhanced compared to the former version, which was presented by Gronz et al. (2016). Furthermore, advanced data post-processing and analysis algorithms were used. These sensor fusion techniques are well established in other disciplines like robotics but must be adapted to geoscientific research objectives, which is - to our knowledge - done for the first time in landslide science within the presented study (although rudimentarily in the first instance). Of course, there are limitations of the current probe and data analysis that must be communicated clearly, which is - again- good scientific practise from our perspective. Beyond that we outlined future improvements, which are already in preparation (e. g. casing modifications allowing larger batteries). (iii) Moreover, the Smartstone

sensor-based probes were successfully used in diverse experimental settings such as single clast transport experiments (Gronz et al., 2016), breakwater experiments (Santos et al., 2019), and rockfill dam overtopping experiments (Ravindra et al., under review).

Resubmission as brief communication.

The presented material fits to the journal objectives that are listed on the journal's web page. One main scope thereby is the presentation of "design, development, experimentation, and validation of new techniques […]". This is exactly what we did in the submitted article. The submitted paper presents substantial scientific results within this scope. From our perspective, it is not possible to identify the research gap, propose an approach to contribute to filling this gap, present an appropriate methodological description, explain the new kind of data for geoscientists, and discuss the technical and scientific results within the format of a short communication. We are aware that not each of these aspects intrigues every reader. But we also belief that it is good scientific practice to draw the whole picture and to document all aspects of a study with an appropriate level of detail, rather to publish individual short reports.

The presented study and consequently the content of the submitted article exceeds the frame of a short communication significantly and must therefore be presented in a research article.

Specific comments:

P1, L2: I am not convinced that every reader understands what is meant by external and internal information.

It would be a solution to describe corresponding meanings at the beginning of the abstract. This will result in a lengthy abstract. From our point of view, the explanation should be given in the introduction, which already contains it.

P1, L4: The first "internal" can be deleted.

We disagree. *Internal* behaviour can – to a certain extend – be inferred from *external* observation. Or we can measure *internal* behaviour by *internal* measurements. Both of which are different approaches and to distinguish between them, both "internal" are needed.

P1, L6: "artificial laboratory-scale landslide" - artificial may be deleted

The word "artificial" was deleted.

P1, L7-10: Is this detailed information really adequate for the abstract?

Yes, it is! If a researcher scans literature searching for an appropriate sensor for his or her own experiment, he or she wants to efficiently exclude the sensors not applicable due to their size right at the beginning in the abstract.

P1, L11: I partly disagree with this statement - the movement of individual pebbles was observed with the Smartstone probe, but not the motion of 520 kg of the pebble material...

This aspect was clarified within the text.

Using the Smartstone probe, the motion of single clasts (gravel size, $d_{50}$ of 42 mm) within approx. 520 kg of a uniformly graded pebble material was observed in a laboratory experiment.

P1, L12: Which mass is meant - the mass of the pebble-material?

The pebble material is meant.

Single pebbles were equipped with probes and placed embedded and superficially in/on the material.

P1, L13-21: This is mainly a description of what has been done - what is lacking is a more generalized description of the results - i.e. the novelty aspect of the study should be better highlighted.

The ending of the abstract has been changed:

Compared to other observation methods Smartstone probes allow for the quantification of internal movement characteristics and, consequently, a motion sampling in landslide experiments.

P2, L28: In my opinion the paper would benefit from additional considerations (including a review) on granular flow mechanics and how these can be described using the sensor data.

The article does not deal with the physics of granular flows. The material has been chosen due to different reasons: to prove the concept of the Smartstone and to allow for multiple repetitions under similar conditions, like described in Sect. 2.4.

P2, L32: What is meant by "some depositional features"? A more general description of the landslide processes would be helpful.

Meant is the extraordinary spreading of very large granular avalanches. This has been clarified in the text:

For instance, Davis & McSaveney (1999) reproduced dry granular avalanches and concluded that the extraordinary spreading of very large granular avalanches may be caused by phenomena like rock fragmentation.

P2, L36: What is meant by precess? Is "process" meant? I am also not sure that I understand what is meant here.

Process is meant. It has been corrected. No change in relative position means that blocks that started in the frontal part of the body stopped at the frontal part of the deposit as well (and vice versa).

P2, L37: Include year for reference Okura et al.

Has been included.

P2, L38: My understanding of the word "collusion" seems to be different from the understanding of the authors. Maybe "collisions" is meant by the authors?

Yes, collision was meant. Has been corrected.

P2, L40: The authors use specific terminology which has not been defined before. As indicated above, a more general description of landslide mechanics and granular flow would be helpful (also for the better understanding of the subsequent passages).

The submitted article is not intended as a review of granular flow experiments or landslide mechanics.

P2, L48: Please specify what kind of 2D section of the body is meant.

Has been clarified.

By means of (high-speed) video analysis such as particle image velocimetry (PIV) or the so-called fringe projection method (Manzella, 2008), only the surface or transversal sections of the body can be analysed.

P2, L51: This is true - but is the information also relevant for the description of the movement of the granular material?

This is correct, but we try to summarize the different types of instrumentation of landslides experiments, and this one should also be mentioned.

P2, L57: Replace "got" by "became"

Has been changed.

P2, L58: What is the technical aspect of so called 'smart tracers' for natural transport? These tracers can be helpful to collect data for the description of natural transport processes but have no effect on natural transport... (please try to be specific language wise throughout the manuscript).

Has been clarified.

Several studies focused on technical aspects (i. e. hardware and software development) of so called 'smart tracers' used to investigate natural transport processes (e. g. Spazzapan et al., 2004; Cameron, 2012).

P3, L59: What is the significance of this sentence?

It should demonstrate the wide range of applications for this method. It has also been clarified within the text.

Others applied these techniques to geoscientific or geotechnical questions, such as the impact of waves on armour units of breakwaters (e. g. Hofland et al., 2018).

P3, L65 and following: This could be more concise.

Has been clarified.

Additionally, the SSP needs wires for energy supply and data transmission and these wires confine a free movement of the device within the soil.

P3, L75: As already mentioned - the physics of the movement of granular material should be better highlighted. I fully agree with the next statement that the manuscript focuses on sensor application (i.e. on the method), and this is exactly why I see limits regarding the significance and novelty of the scientific findings. Therefore I finally recommended to resubmit the manuscript as a brief communication instead of a research paper.

The paper does not try to draw new conclusion on the physics of granular flows. In fact, this type of landslide was chosen as an example to test the usage of the Smartstone probe for experimental landslide science and what additional information can be collected (see also response to Referee #1).

P3, L79: Please improve the description of the objectives.

The description of our objectives has been modified:

1 There has been a significant technical improvement since Gronz et al. (2016) introduced the first version of the Smartstone prototype. Therefore, one objective is is the description of the recent Smartstone probe. In addition, we document major changes to the former version and the corresponding technical specifications.

2. Beyond that, we explain additional information that is supplied by smart sensors and illustrate the specific properties of motion data. Based on a quantitative interpretation, we give an introduction how to read motion data in terms of flume-scale landslide movements.

3. Subsequently, we demonstrate how physical movement characteristics can be derived from the measured raw data and in what way they are different.

4. Further, we highlight the potentials of two- (2D-) and three- (3D-) dimensional visualisation of the paths a clast took during the movement and how these visualisations allow for an easy recognition of complex motion patterns.

5. Finally, we investigate the limitations of the Smartstone prototype and discuss what developments will be necessary to improve the probe and data handling further.

P4, L93: Why "mainly"?

Because high-speed video observation was used as well but the focus of data analysis laid on probe data.

P4, L100: Why "available"?

The word was deleted.

P4, L122: Different dimensions have been mentioned before (L96) which is confusing - I find it also confusing that the dimensions are given only "approximately" - what are the exact dimensions?

As described in the text, the length of the casing can be adapted depending on the size of the plastic plugs that close the tube (see lines 96 to 101). Numbers are given as "approx.", because they may vary slightly as the probe is a prototype and not a standardised device: each tube is manually shortened using a saw.

P4, L124: What is meant by "lager objects"? Is "larger objects" meant? Please check the language throughout the manuscript (I stop here giving comments on the language).

Has been corrected. We will engage English copy-editing services of Copernicus.

P5, L125: Sentence starting at L124 - what is the significance of this sentence for the study?

This information may not intrigue everyone but is intended for researchers who would like to use this technique. We belief it is helpful to be informed about the options of adaption during planning of other experimental campaigns.

P5, L127: This information could be given in the Acknowledgements or incorporated in the above text. The sentence that further improvements are planned already indicates that the presented findings are premature (similar statements are given at the end of the manuscript).

From our perspective it is good scientific practice to indicate that a measuring instrument was developed in cooperation with a private enterprise company and that the device is a prototype, which will be developed further in future. The company is mentioned within the text, because demonstrating the improvement and development process is a key feature of the article.

P5, L137: The reference systems could be introduced better... I find the presentation of Figure 1c rather confusing (as the reference system 'f' is only introduced at L150).

The description has been entirely rewritten:

To compare relative movement characteristics like distance or velocity of different probes, the inner data / coordinate system *p* must be transformed into an outer reference system *rel* (Fig. 1, c). The simplest way to do this is the construction of a reference system using the probe's starting position as coordinate origin. The system is defined by the sensor's inner coordinate system of the first timestep rotated so that z-axis follows gravity. The axes of this relative outer coordinate system are donated with $x^{rel}$, $y^{rel}$ and $z^{rel}$. After the motion has started, the probe's inner orientation will change while the outer reference system keeps its axes configuration. Consequently, within this reference system, it is possible to calculate the probe's orientation and the covered distance in each timestep. In Fig. 1 (c) for instance, the probe has changed its orientation significantly compared to its starting position while moving along the assumed trajectory.

However, the different probe-specific outer coordinate systems must be transformed into the same local reference system to compare different probes' trajectories. For the present study, this local reference system is oriented towards the experimental flume (see section 2.4). Following the former conventions, the axes were donated as $x^f$, $y^f$ and $z^f$. Note that the axes orientations of the outer (rel) and the local reference system (f) may not be identical, except of $z^{rel}$ and $z^f$, as they follow gravity.

In different applications, where a global positioning is required, reference points of the outer coordinate systems must be known in the global system to determine the absolute probe position in the global system.

P5, L140: Why is 'relative' in italics? I don't think that "donated" is adequate terminology... (throughout the paper)

Italics are typically used to emphasize a word. Which is what we wanted to do. Donated should mean denoted. Has been changed (throughout the paper).

P5, L143: Why 'probably'?

Because it is almost certain but not sure.

P5, L151: I don't understand - please improve the description and be more precise (e. g., it is only mentioned below that z follows the direction of gravity).

The description has been rewritten completely, please check response to RC P5, L137.

P5, L153: What is meant by higher-order global system (i.e., why higher-order)?

The description has been rewritten completely, please check response to RC P5, L137.

P5, L154: Why was this not done?

The natural magnetic field is disturbed within the experimental facility due to ferro-concrete surrounding. Additionally, the description has been rewritten completely, please check response to RC P5, L137.

P6, L162: I am not sure that I understand what is meant by (i) as well as (ii and iii).

All these errors result from the sensor technology. We simplified the description to:

The recorded acceleration values of each axis ($a_x^p$, $a_y^p$, $a_z^p$) are generally erroneous due to two reasons: (i) A (quasi-) constant misreading. The mass inside the sensor, which moves to measure acceleration, is not precisely equal in all sensors (manufacturing tolerance), resulting in a bias as well as a linear scaling of true values. (ii) The imprecise orthogonal alignment of the sensor axes and crosstalk. This means that a fraction of each axis acceleration will result in readings at the two other axes.

P6, L163: 'describes' must be 'describe'.

Has been corrected.

P6, L165: All this remains a black box and could be explained in some more detail. I also find that the 'damaged probe' is mentioned too often...

The black box is described in the given reference Frosio et al. (2009). We removed the sentence regarding the damaged probe in this paragraph, as it is not necessary here.

P6, L170: Some more details would be desirable (or at least references where relevant information can be found). I find this important section rather short.

We merged this paragraph with the next one, where the explanation as well as the reference are given.

P6, L178: What is meant by 'various studies'?

Future studies are meant.

The design of the experimental setup focussed on an exact and rapid triggering mechanism of the artificial landslide and flexibility for future studies.

P6, L180: A more precise length is given in the figure, which I find confusing (why approx. when the exact length is known?).

We changed the number to 4.24 m.

P6, L181: I find this difficult to read - please improve language.

Has been improved.

Some clasts also reached the lower part of the flume, which is inclined by 10 °.

P7, L206: Please check language. What was the accuracy of these measurements?

Has been changed:

This was done using a laser distance meter (accuracy +/- 1 mm).

P7, L210: This statement defines the scope of the study - the demonstration of the applicability of the sensor to monitor the motion of individual pebbles within a landslide - this calls again for publication as a brief communication instead of a research article.

As argued above, presenting the scientific progress (technical, analytical) of this technique since 2016 is not possible in the form of a brief communication.

P7, L211: The information on the date should be presented in Section 2.4 (is this information important)?

Has been deleted.

P7, L216: As mentioned above, the calibration of the sensor remains partly foggy, and I am not sure that the terminology "calibrated raw data" is adequate - it seems to me that raw data are presented which were obtained by a calibrated sensor?

The calibration description has been modified as written at comment concerning P6, L162. Details are described in the given reference. The word "raw" has been removed. This correction has been done in the complete article.

P7, L217: Could this be measured without activated IMU? If not, the statement in brackets could be deleted.

Has been deleted.

P7, L217/218: I had first difficulties to understand this sentence when looking at the figure - this could be formulated more clearly.

Has been clarified.

Note that the three curves of $x^p$, $y^p$ and $z^p$ (Fig. 3, a) show the acceleration along the particular axis (see below). The gyroscope data curves (Fig. 3, c) show rotation around these axes. At the top of each plot, white bars indicate stationary (no motion) and black bars non-stationary (motion) periods.

P8, L222: Why does the time series begin at a 'negative time'? When exactly was the landslide triggered? This should be mentioned/explained more clearly. Also, the statements relate to accelerations along the considered axes, this should be better highlighted in the text (xp and yp show low values - but strictly speaking, these are coordinate axes; the same applies to the statements given in the next sentences).

Negative time has been clarified:

The start of motion of pebble 4 was defined as 0.0 s.

All statements related to "axis shows values" have been changed to "values along this axis".

P8, L223: Why 'seems'? The data show this higher level.

Has been changed.

Before the actual motion begins (stationary conditions, left white bars in Fig. 3), low values were recorded along $x^p$ and $y^p$, though $x^p$-readings are on a slightly higher level (approx. 0.0 g).

P8, L230: Please improve - this is difficult to read and understand (suggestion: alpha, beta and gamma define the angle between xp, yp and zp and the gravity vector).

Suggestions are thankfully accepted.

[…] where α, β, and γ give the angle between $x^p$, $y^p$, and $z^p$ and the gravity vector, respectively.

P8, L230 - L312: This qualitative presentation of the raw data is difficult to read and understand - it should, in my opinion, be coupled with the derived trajectory.

This section is intended to be an introduction on how this kind of data has to be read and interpreted. We still see the need of a detailed explanation, what motion features are displayed by the data. The paper is structured from low complexity to higher complexity (single clast raw data - single clast derived data - single clast spatiotemporal visualisation - multi clast spatiotemporal visualisation). From our perspective, explanation and understandability of the content would not benefit from the mixing of these complexity levels.

P8, L231: 'downwards direction' is, in my opinion, a rather confusing terminology.

Has been changed.

Accordingly, under static conditions the probe's orientation relative to the gravity vector (vertical direction) can be calculated from the three readings of $a_x{}^p$, $a_z{}^p$ and $a_z{}^p$.

P8, L234: If the pebble is stationary, it is clear that it cannot rotate...

Constant readings (except for noise) do not necessarily indicate stationary conditions, as we always have non-zero readings due to gravity. The probe might also move. If also the gyroscope indicates that no rotations occur, the conclusion that the probe is not moving becomes more probable. It is correct that this sentence should be placed above. The new sentence in line 224 is:

This pattern represents non-motion conditions, where only gravitational acceleration is recorded. This assumption is supported by the zero readings of the GYR.

P8, L234: I guess the landslide was triggered at 0.00 s? This should be mentioned in the text. Define 'the change' in the plots more clearly.

This has been explained before (see comment P8, L222) and is also clarified within the text.

A sudden change in the axes-readings at 0.0 s is visible in all three plots.

P8. L235: The text implies that the zp-axis drops, which is not the case (and which is rather confusing). What drops is the acceleration value, and this needs to be described more clearly (also in the following passages).

Has been clarified.

Between 0.0 s and approx. 0.03 s, a clear drop of $z^p$-recordings to the halve of the former level is visible in the acceleration plot (Fig. 3, a).

P8, L238: This could be explained better and the motion characteristics 'free fall' and 'hampered free fall' should be defined more clearly.

Has been clarified.

Low absolute values of acceleration can only be achieved if free fall (unconfined acceleration within the earth's gravitational field into the direction of its centre of mass) is mixed with an additional. Thus, values between 0 g and 1 g imply a hampered free fall (no completely developed free fall, confined motion) and/or an additional lateral acceleration.

P9, L248: This seems to be rather speculative - a statement like 'a conclusion might be...' is very vague...

Has been changed.

We conclude that the surrounding part of the mass moves coherently downwards.

P9, L251: Note that the values range between 1 g and -1 g (and do not correspond to 1 g and -1 g as presently stated).

We mixed up the letters in the description. Correct description:

Along $y^p$, values around -1 g were recorded; along $z^p$, values around 1 g were recorded.

P9, L257: However, I can see other spikes which are defined by two data points.

Therefore, we wrote "most of the other peaks". These features indicate short periods of pronounced acceleration instead of single hits, which is worth mention. Moreover, from these observation distinct motion patterns can be interpreted as demonstrated within the text.

The strongest peak of the whole sequence (approx. 4.6 g) is measured at $z^p$ for two subsequent readings. Thus, the change in velocity is bigger than all other changes as the strongest absolute acceleration also lasts longer than most other acceleration peaks, which only consist of one reading.

P9, L260: Indicate which axis is analyzed here.

Has been clarified.

Here, the phase begins with relatively low $\omega$ of approx. 260 ° $s^{-1}$ at 0.898 s on $y^p$.

P10, L289: 'Of cause'?

Has been corrected.

P10, L290: Why?

Because we know that the flume is oscillating after the impact of the material. We clarified the text:

Concerning the whole time series, some interesting aspects shall be mentioned: The small deviations from the mean axes readings of the ACC after the motion (right white bar) can be interpreted as oscillation of the flume construction after the impact. This is supported by the pattern: uniform oscillations with gradually decreasing amplitude.

P10, L290: This is trivial - I would be surprised if the pebble would not change its orientation.

Yes, one can see and would also expect that the orientation will change during the motion. Apart from that, one can interpret this from the data. Consequently, this interpretation could be done as well if the object would not be visible (e. g. embedded into the material).

P10, L313: The above qualitative statements regarding the motion mode should be coupled with the considerations of the movement with respect to position and time.

Has been changed.

This is also supported by the alternating pattern of high $a$ peaks and almost zero acceleration magnitude. This pattern results from saltation as the pebble bounces at the flume bottom before it rebounds and falls again.

P10, L315: This could be mentioned earlier.

It is mentioned there, because the statement serves as a bridge passage to the quantitative movement analysis.

P11, L316: Are these relationships really that simple?

Simple has been replaced by basic.

P11, L317: Check language ('calculated after...').

Has been changed:

[…] according to Eq. 1 […]

P11, L318: Why were the angles 'received'? How can the accelerations be 'rearranged'? Please check language throughout.

"Received" has been changed to "calculated". "Rearranged" has been changed to "rotated".

P11, L320: This is a description of a method which is described in detail in the literature - it could therefore be included into Section 2.

The methodological aspects are described at this part of the text, because it is one of the key features of the study that we apply these algorithms on geomorphic motion data, which was not done before. Again, this is also why we describe it in detail.

P11, L328: 'srel' is not integrated. It is obtained from integration.

Has been changed.

After the rearrangement of the recorded accelerations and with respect to time t, the movement characteristics $v^{rel}$ and $s^{rel}$ can be obtained from the integration

P11, 336: I am not sure that I understand what is meant here. Please improve.

Has been changed.

During stationary periods these values are defective. This can be seen at $x^{rel}$ (Fig. 3, a), where values of approx. -4 m s$^{-2}$ were calculated. Obviously, this cannot be true as the pebble does not move. However, these false calculations are excluded from further integration (compare Fig. 3}, b and c) and do not influence the following interpretations.

P11, L337 and following: All this could be combined with the qualitative discussion.

The paper is structured with increasing degree of complexity. Mixing different levels would not be beneficial (see response to comment P8, L230 - L312).

P11, L338: I disagree with this statement: a_zrel does not increase but decrease from 0 to -9.4. It also does not do this continuously decrease as clearly shown by the fluctuations in the plot.

This important aspect was already discussed within the text. An acceleration reading on a particular axis represents the magnitude of acceleration along this axis. The positive or negative algebraic sign refers exclusively to the direction. It does not imply an increase or decrease. Consequently, if the values change from 0 g to -9 g, acceleration increases (the object is accelerated more) against the direction this axis is pointing to.

We agree that "continuously" is the wrong description.

As displayed in Fig. 4 (a), $a_z^{rel}$ generally increases until at approx. 0.32 s a local maximum of approx. -9.4 m s$^{-2}$ occurs.

P11, L339: Why? This should be explained better. I also cannot see in the Figure that the pebble 'swims'. Please improve.

This aspect was explained before for acceleration in "g"-units. The explanation only has to be adapted to the other unit. Only if earth's gravitational acceleration is calculated as resultant acceleration magnitude,

pure free fall conditions are present. If the calculated resultant acceleration magnitude is ≠ 9.81 m s$^{-1}$, other motion components (or the underlying material, for instance) confine a free fall.

The verb "swim" has been replaced by "moves".

P11, L342: What is the significance of the maximum velocity v_yrel? Why is v_zrel decreasing afterwards while it further increased simultaneously? This is confusing.

The variables $v_y^{rel}$ and $v_z^{rel}$ were swapped. This has been corrected.

P11, L343: Why 'covered'?

"Covered" has been replaced by "calculated".

P11, L345: I disagree again - the plots for the y-axis do not really exhibit a major change. I am also not sure about the relevance of the capture of the high-speed sequence.

Modified text:

At 0.389 s after the start, a discontinuity at x- and z-axes is visible in fig. 4 (a) and (b).

P11, L346: Is the 'acceleration curve' really smooth before 0.389 s?

No, it is not. Modified sentence:

The variability of the acceleration time series increases.

P12, L349: This is difficult to understand and should be explained better.

If no hits would be present between the clasts, no acceleration peaks would occur. This means: a peakier curve indicates pronounced contacts. However, it is not necessarily friction but, more general, energy dissipation. New sentence:

Additionally, the peaky pattern of the acceleration and velocity curves indicates pronounced clast contact and energy dissipation.

P12, L351: What is the significance of this observed maximum?

It is the maximum velocity for this probe.

P12, L353: Please explain better - this is hard to see from Figure c.

We added a detail figure in Fig. 4 (c).

[Figure]

P12, L355: Figure 4 d shows photographs and not displacement plots?

This was a wrong cross reference. It has been corrected to Fig. 4, c.

P12, L355: I do not see these staircases (if plot c was meant).

See detail figure two comments above.

P12, L372: This was mentioned before.

The sentence was removed.

P13, L390: I am not sure that I understand what is meant.

The sentence has been modified:

On the side view plot (Fig. 5, a), $y^p$ and $z^p$ are almost drawn in full length, whereas $x^p$ is short, indicating its orientation towards the viewer's perspective.

P13, L390 and following: This could be combined with the previous analyses.

Again, the paper is structured with increasing degree of complexity. Mixing different levels would not be beneficial (see previous responses).

P14, L432: This sentence is incomplete.

Has been corrected.

It is visible that pebble 1 and 2 were embedded into the material, whereas pebble 3 and 4 were placed at the surface of the material (see Fig. 2).

P14, L437: This indicates that further work is required.

From our point of view, it is common and good scientific practice to communicate advantages as well as deficiencies that have to be dealt with if a method is presented to the scientific community. Nevertheless, further improvements are necessary. This will always be the case if scientific progress is sought. Apart from that, necessary improvements are discussed in detail within the text. This includes also comments for benefits that are expected from particular improvements.

P15, L470: This is a drawback for field applications, but not really for laboratory experiments.

This might also be problem under laboratory conditions. For instance, if the probe is built within a model slope that will fail due to artificial rain. Here, the time of failure is not known, but batteries cannot be changed. This issue is currently the objective of our work.

Additional References:

Santos, J. A.; Pedro, F.; Coimbra, M.; Figuero, A.; Fortes, C. J. E.M.; Sande, J.; Körner, M.; Lemos, R.; Bornschein, A.; Weimper, J.; van den Bos, J.; Dost, B.; Hofland, B.; Carvalho, R. F.; Alvarellos, A.; Peña, E.; Pohl, R.; Kerpen, N. B.; Reis, M. T. (2019): 3-D Scale Model Study of Wave Run-Up, Overtopping and Damage in a Rubble-Mound Breakwater Subject to Oblique Extreme Wave Conditions. In: Defect and Diffusion Forum 396, pp. 32–41. DOI: 10.4028/www.scientific.net/DDF.396.32.

Ravindra, G.; Gronz, O.; Dost, B.; Sigtryggsdóttir, F. (under review): Description of failure mechanism in placed riprap on steep slope with unsupported toe using smartstone probes. In: Engineering Structures.

---

## Author Response (AR2)

**Response to Referees after Major Revisions**

Referee #1 text is grey.

Referee #2 text is black.

Response text is blue.

Changes are red.

**Referee #1:**

The authors did improve the MS following some of my comments, but many of the comments were not accepted by the authors or they answer to me as reviewer but did not improve the MS. I thus recommend on improving the MS, and I recommend reconsidering after major revisions. Some examples - L105, L223, Lee5, L437, L510.

There were roughly 100 specific comments from both reviewers. We are thankful for taking the time to give that many specific comments in a review. Roughly 75 of the comments resulted in changes in the document. The amount of changes becomes visible in the version with changes marked. However, there were approx. 25 comments that did not result in changes. For each one of them, we provided a detailed explanation why we did not change the text accordingly. From our point of view, we cannot include every answer to all comments in the document, because of the limited length of the text and the fact that too many details would impair the readability of the MS. Nevertheless, besides the explanation in the point-to-point answer, a completely new section was added to the article (Sect. 4.4, see Appendix). It contains additional information.

Detailed answers to all mentioned comments are given in the appendix at the end of this document.

The two first sentences of the abstract are not clear, and I do not see why the authors did not improve and clarify them.

We improved the first paragraph as follows:

Laboratory landslide experiments enable the observation of specific properties of these natural hazards. However, these observations are limited by traditional techniques: frequently used high-speed video analysis and wired sensors (e.g. displacement). These techniques lead to the drawback that either only the surface and 2-dimensional profiles can be observed, or wires confine the motion behaviour. In contrast, an unconfined observation of the total spatiotemporal dynamics of landslides is needed for an adequate understanding of these natural hazards.

We think the first version does indicate the difference between "conventional" and the Smartstone techniques in a better way.

Implications out of the lab and potential implications to understand landslides.

We illustrated the potential applications of the Smartstone probes for geoscientific and engineering disciplines in Sect. 4.5. We added the following paragraphs:

In the future, Smartstone probes may help to explain observations from the modelling of landslide motion processes. For instance, it would be interesting to investigate the influence of clasts with different sizes. Phillips et al. (2006) observed a uniform distribution of fine and coarse particles in

laboratory high-mobility granular flows. Providing the right scale (compare Sect. 4.4), trajectory reconstruction of different clasts may shed light on the question how different clast sizes segregate during the transport process. A deeper analysis of the probe data may also allow for estimations of energy dissipation within the landslide body during the motion process (compare e.g. Manzella & Labiouse, 2013).

Beyond these geoscientific objectives, the Smartstone probe was already used in experiments focussing on coastal- and hydro-engineering problems. Santos et al. (2019) briefly reported experiments to investigate the stability of breakwater amour units. By means of the Smartstone probe data, Ravindra et al. (2020) presented a detailed analysis of the failure mechanism of placed riprap on laboratory dam models. These examples demonstrate the broad applicability of the Smartstone technique.

**Referee #2:**

Technical Note discussion

With the detailed explanation provided in this second review, the reviewer's point of view becomes clearer for us: We can follow the arguments to change the article to a technical note. From our point of view, this should be the editor's decision. However, shortening the article will not be as easy as proposed in the review, as both reviewers asked in their first review for more details and explanations, which made the revised article even longer. Shortening would (also) affect some of the extra content both reviewers explicitly asked for.

Probe version v2.1 was used in recently published study (Ravindra et al., 2020). Contrary, for the study that is presented here, v2.0 was used.

The detailed analysis that is presented within the submitted manuscript was developed and used for the first time with the data that was recorded with v2.0. Meanwhile, the probe itself was further developed. In v2.1 a firmware update and some error fixes were done. In addition, the GUI of the operating software was slightly changed. This illustrates the previously mentioned "further development". Therefore, it was possible to use the v2.1 for the experiments published by Santos et al. (2019) and recently by Ravindra et al. (2020). These examples are now mentioned within the text. Probe version v2.0 and v2.1 have very similar technical specifications. Therefore, all data sets have the same structure.

Detailed abstract.

We still think that this information is important for researchers that use similar techniques or are interested to use similar probes for their experiments.

**L14:** Please check use of tense (compared to the previous sentence).

Changed:

In a first analysis step, the data of one pebble is interpreted qualitatively, allowing for the determination of different transport modes, such as translation, rotation and saltation. In a second step, the motion is quantified by means of derived movement characteristics:

**L41:** Please check the sentence starting in this line.

Changed:

This contrasting initial condition was used as an indicator for fragmentation.

They found that the potential internal and external friction strongly influence the energy dissipation during the displacement process.

**L130:** I still think that such a sentence is not adequate here and should be located at the end of the paper in the conclusions or outlook.

We shifted this statement to the end of the paper. Extraction from the last paragraph:

The present study also demonstrates the potentials of cooperation between private enterprise companies and research institutes. The Smartstone probe was developed and manufactured in cooperation with the company Smart Solutions Technology GbR, Germany.

**L207:** This has been mentioned before (e.g., L75 and L95).

Changed:

Hereinafter, Smartstone probe data of one experiment was chosen to present (i) sensor recordings (Fig. 3), (ii) the derived movement characteristics ($a$, $v$, $s$, Fig. 4), and (iii) 2D- and 3D-visualisations (Fig. 5, Fig. 6).

**L297:** But isn't the pebble most of the time "somewhat tilted"?

Clarified:

Because the stationary ACC readings of $z^p$ are slightly lower, it follows that this axis does not point exactly into vertical direction after the motion. The probe is oriented in a different way than prior to the experiment.

**L314:** Replace "Now, we want to" by, e.g. In the following, the movement is investigated with respect to..."

Has been replaced:

In the following, the movement is investigated with respect to position and time.

**L525:** I still infer that the question why e.g. Kalman filtering or Markov-localization has not been done in the study.

Because applying these mathematical techniques on complex motions (more complex like the human gait in terms of motion pattern) is a difficult task and object of recent research.

**L578:** Redundant information (see L69).

Has been deleted and changed:

Beyond the derivation of Euler angles, the present study demonstrates that the calculation of movement characteristics and the reconstruction of spatiotemporal trajectories are essential to describe geomorphic motion processes adequately.

**Appendix**

The following paragraphs are extracted from the former author's response to the comments of referee #1. We identified about 20 aspects mentioned by referee #1 (including general comments and figure comments) that had to be answered. 13 of them resulted in changes within the manuscript. Below, specific comments and answers are listed that did not result in text changes. Therefore, our previous answers and discussions are given. Additionally, we added some new answers. Apart from that, many red and blue coloured text passages indicate the improvements we applied.

Referee #1 text is grey.

Former response text is blue.

*Additional response text is blue italic.*

Changes are red.

L79 - what is the former version? – maybe I missed.„

The former prototype version is now mentioned earlier in the text. The reference is Gronz et al. (2016).

*The reference is now indicated in L78. We apologise for the missing indication of the change:*

Thereby, methodological and technical progress compared to the former probe version, presented by Gronz et al. (2016), will be demonstrated.

L105 - The sampling rate (100 Hz) is low. Most probably, it cannot record the sharp impulse during a collision with another rigid body. Miss recording such impulse can lead to significant errors in the velocity and position, calculated by integrating the recorded acceleration. The authors should address this issue, provide an estimate of a typical collision duration in their experiment, and show that it is longer than 1/sampling rate. In case the above condition is not fulfilled in the experiment, the authors should explain the implications. A short discussion on the sampling rate in a broader context of a real landslide can be illuminating for the reader.

During the data analysis we did not notice any issues resulting from the sampling rate. Generally, the needed frequency does not depend on the duration of accelerations but on the uniformity of the movement. The sensor does not miss single strong acceleration peaks shorter than the sampling frequency due to its principle of operation: Inside the chip is a small movable piece shaped like a comb (for each axis). It is positioned inside a stationary second comb. Lateral movement due to accelerations changes the distance between the two combs, resulting in an electric capacity change, which is measured to derive acceleration. If a short, abrupt acceleration occurs, the movable comb will be displaced in any case, even if the acceleration does not last as long as the sampling period, due to its inertia. The chip integrates inherently during the sampling period. However, if the sensor is moved several times forward and backwards within a sampling period, the sensor will miss this information, as the movement's frequency is higher than the sampling frequency. Thus, the correct movement cannot be observed. Again, preliminary tests facilitate choosing the correct sampling frequency.

*This comment is about the possible missing of peaks. We provided a detailed explanation why accelerometers of this type do not miss peaks shorter than their sampling frequency. In the new section 4.4, we also discussed that the sampling frequency depends on the movement changes within a moving landslide. The new section is as follows:*

4.4 Scaling

A scaling of the recording ranges will be necessary if the Smartstone method is adapted to other experimental scales or velocities. Additionally, the scaling of temporal persistence of movements has to be respected as the Nyquist frequency to observe the motion without undersampling changes with the rate of movement changes (Yang et al., 2009). This means that a small pebble in a fast-moving landslide will show more abrupt changes in its velocity, trajectory and mode than a large block in a slow landslide. Thus, the ranges of the sensors and the sampling frequency have to be adjusted depending on the landslide velocity *and* the particle size. Several aspects concerning the sensor recording range for different experimental applications have to be considered:

Acceleration range: The expected acceleration depends on the velocity of the landslide, as the strongest peaks occur during nonelastic collisions of moving particles with stationary boundaries, e.g. bedrock. Thus, the range needs to be increased (by choosing a different accelerometer chip in the Smartstone) with velocity. However, to choose a gratuitously large range to avoid clipping is counterproductive, as the quantisation error will also increase, as there is only a limited number of steps within the range. A deliberated balance needs to be chosen, e.g. by performing preliminary tests.

Gyroscope range: The rotational velocity depends on the movement of the landslide but also on the size of particles. For instance, if the mass moves with $1\,\mathrm{m\,s^{-1}}$, a single rolling pebble with 30 mm diameter will show a rotational velocity of $3820\,\mathrm{°s^{-1}}$ (pebble circumference 942 mm, thus 10.6 rotations per second). Thus, the expected range can be calculated using the shortest circumference of the Smartstone's host particle and the expected landslide velocity. Again, choosing a gratuitously large range to avoid clipping will increase the quantisation error.

**L124** - needed?

This information demonstrates that the Smartstone probe can be adapted to different research objectives.

*This comment is due to the mentioned variability in probe size. We believe it is an important information - maybe not for every other researcher - but for some using similar techniques.*

**L223** - Can you clarify 0.0 g means?

Within the gravitational field of the earth, zero g can only be achieved if and only if an object moves in pure free fall without aerodynamic drag. In this case, the resultant acceleration vector of motion and the acceleration vector due to earth's gravitational acceleration exhibit the same magnitude. These vectors, however, point in the opposite direction compensating each other. This results in a measured acceleration magnitude of 0.0 g at a measuring device, such as the Smartstone probe.

*We also adopt some changes in Sect. 3.1 and give a better explanation. This is also indicated by blue and red font colours in the marked-up-version.*

**L225** - The whole section (3.1) is very long and tedious. The formulas are trivial, in any case, the authors do not use the projections of g in the discussion.

We expect different subsets of so-called trivial knowledge at different readers. Thus, this section is an introduction on how this kind of data has to be read and interpreted. In discussions with colleagues it emerged that it is often not clear what is displayed within the graphs (e.g. Fig. 3). To our knowledge, there is no publication that describes spatiotemporal motion data adequately in a geoscientific context. Therefore, we decided to give a detailed description in this article.

*We think this comment is meant by "Lee5". We justified our decision to give a detailed explanation. The mentioned formulae are given to demonstrate the basic laws that are used to calculate the movement characteristics to a geoscientific reader.*

**L437-442** - The most probable reason for the wrong trajectory of pebble 3 is miss recording of collision with another pebble due to the slow sampling rate of the used IMU.

As explained above (comment L105), missing an acceleration peak is not possible due to the design of the acceleration sensor (as long as the motion frequency does not exceed the sampling frequency, which is given in this case). Therefore, this cannot be an explanation for the erroneous inclination of the trajectory. In addition, an overestimation of the vertical trajectory component occurs right from the beginning of the motion. The peak would have to be occurred before the motion starts, which obviously is not possible (not least because pebble 3 was placed at the surface of the material).

*We answered to this comment in detail and discussed the explanation. From our point of view miss recording of collision is not the reason for this deviation.*

**L510** - In the present study, the probe monitored movements over a short period of ~ 2 sec. A brief discussion regarding the expected error in retrieving the trajectory over more extended periods can help to assess the type and scale of landslides that can be monitored in this way.

We agree that the duration of recorded motion is a critical aspect of this technique. Because the estimation of the displacement components requires double-integration, absolute errors will increase with time – depending on the motion. Therefore, longer durations will generally result in less accurate absolute results. The relative error will remain stable for certain kinds of movements. However, a general extrapolation of the expected error is not trivial, as it depends on the ratio of noise, quantisation error etc. to the magnitude of the true accelerations.

*We apologise that we indicated the wrong referee by the black font colour instead of grey. Apart from that, we answered and discussed the comment.*

[revised manuscript text omitted]